# The Labyrinth and the Thread: Rethinking Regularizations in Sequential Knowledge Editing for Large Language Models

**Zheng Wang** [1 2]  **Kaixuan Zhang** [1 2]  **Wanfang Chen** [1 2]  **Jingwen Zhang** [2 3]  **Xiaonan Lu** [1 2]

## Abstract

Sequential editing of structured knowledge in large language models allows targeted factual updates without retraining, yet existing methods often rely on complex regularization or constraint mechanisms whose necessity remains unclear. In this work, we systematically investigate the mechanisms underlying effective and stable sequential editing. Specifically, we first analyze the empirical success of AlphaEdit and establish, via a rigorous optimization analysis, the formal equivalence between one-time and sequential editing. Building on this insight, we generalize the equivalence to a broader class of editing objectives, demonstrating that stability emerges naturally from properly accounting for accumulated editing constraints, rather than from specialized regularization or null-space operations. We empirically confirm that many commonly used regularization strategies are unnecessary for reliable sequential updates. Furthermore, we extend our framework to handle conflicting edits, ensuring robust and consistent behavior under contradictory updates. Ultimately, our work provides Ariadne's thread through the labyrinth of sequential editing, charting a path toward simpler, more interpretable, and dependable knowledge updates. Our code is available at https://github.com/Wangzzzzzzzz/OTE-SE-Alignment.

## 1. Introduction

Large language models (LLMs) increasingly serve as repositories of factual knowledge, much of which can be represented as structured triples of the form (subject, relation, object) (Meng et al., 2022). As these models are deployed in dynamic and evolving environments, the ability to precisely update or correct specific structured knowledge (Sinitsin et al., 2020) without retraining from scratch has become a central challenge. This has spurred the growth of structured knowledge editing, a field dedicated to modifying a model's behavior on targeted factual triples while preserving its general capabilities (De Cao et al., 2021; Wang et al., 2024; Hu et al., 2025).

Among existing approaches, parameter modifying methods based on the locate-and-edit paradigm have been particularly influential (Meng et al., 2023). These methods first identify parameters associated with a target fact and then apply constrained updates to induce the desired change. To enable sequential or lifelong editing, recent methods incorporate a variety of mechanisms, including null-space projections in AlphaEdit (Fang et al., 2025), implicit regularization via post-processing operators (Gu et al., 2024; Ma et al., 2025), and constrained optimization objectives (Meng et al., 2022). Collectively, these designs aim to mitigate interference between edits and preserve previously edited knowledge. Notably, AlphaEdit (Fang et al., 2025) stands out for its strong empirical stability in sequential editing. However, the growing diversity and complexity of these mechanisms also raise a fundamental question:

> *What are the essential ingredients that ensure successful and reliable sequential model editing?*

Despite some prior attempts to address this question, existing studies (Li et al., 2024; Gupta et al., 2024; Li & Chu, 2024) are either primarily empirical, lack a unified theoretical foundation, or fail to fully explain the empirical success of methods like AlphaEdit (Fang et al., 2025). In this work, we aim to provide a principled answer by addressing the following research questions:

**RQ1:** To what extent does AlphaEdit's empirical success stem from its null-space projection mechanism, and how critical is this component in practice?

**RQ2:** Beyond AlphaEdit, what general principles govern

---

[1]Bosch Center for Artificial Intelligence (BCAI). [2]Bosch (China) Investment Ltd. [3]School of Statistics, East China Normal University. Correspondence to: Zheng Wang <david.wang3@cn.bosch.com>, Kaixuan Zhang <kaixuan.zhang@cn.bosch.com>.

*Proceedings of the 43rd International Conference on Machine Learning*, Seoul, South Korea. PMLR 306, 2026. Copyright 2026 by the author(s).

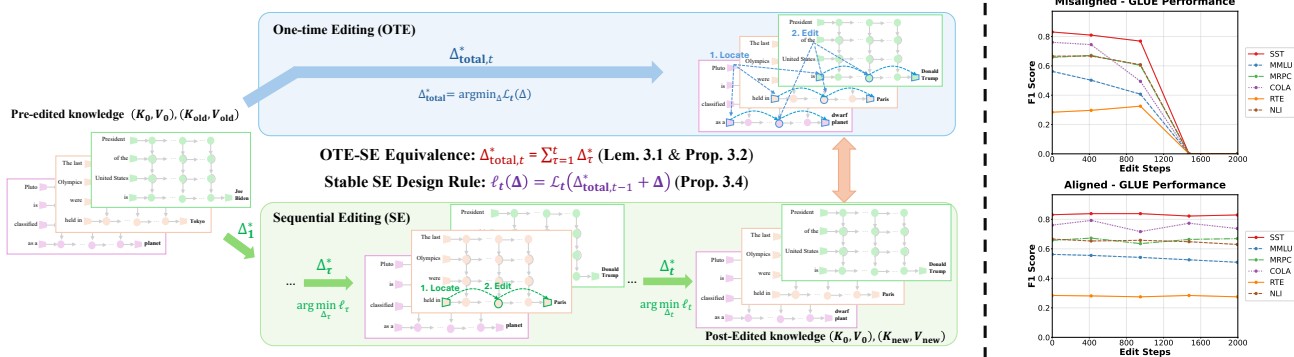

*Figure 1.* (Left) Theoretical insight: equivalence between one-time editing (OTE) and sequential editing (SE) is the key to stable sequential updates; (Right) Empirical evidence on GLUE benchmark: OTE-aligned SE maintains performance, whereas OTE misalignment causes substantial degradation.

effective sequential editing, and can they be distilled into a theoretically grounded design criterion?

**RQ3:** What is the actual impact of complex regularization strategies when applied on top of an already stable and reliable model update? Are they still beneficial?

Answering these questions requires looking beyond individual mechanisms and regularization tricks. Our work shows that much of the apparent complexity in sequential editing is illusory. Beneath the labyrinth of regularization strategies and sequential update rules lies a surprisingly simple and unifying structure. Specifically, we demonstrate that a broad class of locate-and-edit methods can be formulated as ordinary-least-squares (OLS) problems, often admitting closed-form solutions. Within this unified view, sequential editing and one-time editing are mathematically equivalent when past editing constraints are properly incorporated. This equivalence reveals that stability across edits is not an emergent property of specialized regularizers or null-space operations, but a direct consequence of solving the same underlying optimization problem while properly accounting for accumulated constraints. Building on this insight, we further extend the unified formulation to handle competing or contradictory edits, where naive sequential updates often fail. More broadly, our work provides the community with a principled framework for understanding sequential editing, clarifying which mechanisms are essential and which are superfluous. This perspective lays the groundwork for designing simpler, more reliable, and more interpretable methods for updating knowledge in large language models.

## 2. Preliminary

### 2.1. Auto-regressive Large Language Models

A decoder-only large language model (LLM) updates its internal representations through a stack of transformer layers. At each layer, the next hidden state is formed by adding the previous residual stream to the outputs of the self-attention and feed-forward network (FFN) modules. Since prior work has shown that FFN modules play a central role in storing factual associations (Dai et al., 2022; Geva et al., 2021; Meng et al., 2022), we focus on the FFN output projection. For simplicity, throughout the following sections we write $\mathbf{W} \in \mathbb{R}^{d_o \times d_i}$ for the layer-specific matrix $\mathbf{W}_{\text{out}}^{(l)}$, where $d_i$ and $d_o$ denote the input and output dimensions of the FFN layer. Concretely, the layer update can be written as

$$h^{(l)} = h^{(l-1)} + a^{(l)} + m^{(l)},$$
$$m^{(l)} = \mathbf{W}_{\text{out}}^{(l)} \, \sigma\left(\mathbf{W}_{\text{in}}^{(l)} \, \gamma\left(h^{(l-1)}\right)\right),$$

where $\gamma$ denotes layer normalization, $\sigma$ is a non-linear activation function, and $\mathbf{W}_{\text{in}}^{(l)}$ and $\mathbf{W}_{\text{out}}^{(l)}$ are the FFN weight matrices.

### 2.2. Structured Knowledge Editing

A growing body of evidence suggests that structured factual knowledge, typically expressed as subject–relation–object triples $(s, r, o)$, is predominantly stored in the FFN layers of large language models (Meng et al., 2022). This perspective naturally motivates the locate-and-edit paradigm, where targeted knowledge updates are performed by directly modifying $\mathbf{W}$ rather than retraining the entire model. Conceptually, each fact can be interpreted as an implicit key–value association: the hidden representation of the prompt $(s, r)$ acts as a key $k$, while the FFN output leading to the object $o$ serves as the corresponding value $v$. Updating a fact thus corresponds to introducing or adjusting these key–value associations through careful parameter modifications.

To update a fact, we construct key–value pairs $(\mathbf{K}_{\text{new}}, \mathbf{V}_{\text{new}})$ representing the desired new knowledge. We seek an optimized perturbation $\mathbf{\Delta}^*$ to the weights $\mathbf{W} \in \mathbb{R}^{d_o \times d_i}$ such that $(\mathbf{W} + \mathbf{\Delta}^*)\mathbf{K}_{\text{new}} = \mathbf{V}_{\text{new}}$, thereby incorporating the new associations. Meanwhile, we preserve existing knowledge

captured by a set of key–value pairs $(\mathbf{K}_0, \mathbf{V}_0 = \mathbf{W}\mathbf{K}_0)$.

Combining both objectives, the editing problem can be written as a single ordinary-least-squares (OLS) problem, whose solution admits the following closed form[1]:

$$\begin{aligned}\boldsymbol{\Delta}^* &= \arg\min_{\boldsymbol{\Delta}} \|(\mathbf{W} + \boldsymbol{\Delta})\,[\mathbf{K}_0 \mid \mathbf{K}_{\text{new}}] - [\mathbf{V}_0 \mid \mathbf{V}_{\text{new}}]\|_F^2 \\ &= (\mathbf{V}_{\text{new}} - \mathbf{W}\mathbf{K}_{\text{new}})\mathbf{K}_{\text{new}}^\top \left(\mathbf{K}_0\mathbf{K}_0^\top + \mathbf{K}_{\text{new}}\mathbf{K}_{\text{new}}^\top\right)^{-1}.\end{aligned}$$

### 2.3. From One Time Editing to Sequential Editing

While the classical locate-and-edit paradigm typically assumes that knowledge updates are applied in a single batch, real-world deployment of LLMs often necessitates sequential editing, where the model continuously incorporates new or evolving facts. Compared to one-time updates, sequential edits pose non-trivial challenges, as naive incremental editing strategies may struggle to preserve consistency across edits and maintain overall model behavior (Hartvigsen et al., 2023; Gu et al., 2024; Ma et al., 2025; Fang et al., 2025; Hu et al., 2024). A notable advance in this direction is AlphaEdit (Fang et al., 2025), which enables thousands of stable sequential edits by leveraging sophisticated null-space projection techniques.

Formally, let $\mathbf{K}_0$ denote the key matrix corresponding to the preserved knowledge, and let $\mathbf{P}$ be the orthogonal projector onto its left null space. At editing step $t$, $(\mathbf{K}_t, \mathbf{V}_t)$ denotes the key–value pair associated with the newly injected knowledge. We also define $\mathbf{K}_{1:t-1} := [\mathbf{K}_1 \mid \cdots \mid \mathbf{K}_{t-1}]$ as the concatenation of all previously edited keys, and the residual of the current edit as $\mathbf{R}_t := \mathbf{V}_t - \mathbf{W}_{t-1}\mathbf{K}_t$, where $\mathbf{W}_{t-1}$ represents the model parameters before step $t$.

Under this notation, AlphaEdit performs sequential editing by updating the model parameters additively as

$$\mathbf{W}_t = \mathbf{W}_{t-1} + \boldsymbol{\Delta}_t^*. \tag{1}$$

At each step $t$, the update $\boldsymbol{\Delta}_t$ is computed in closed form as

$$\boldsymbol{\Delta}_t^* = \mathbf{R}_t\mathbf{K}_t^\top\mathbf{P}\left(\mathbf{K}_{1:t-1}\mathbf{K}_{1:t-1}^\top\mathbf{P} + \mathbf{K}_t\mathbf{K}_t^\top\mathbf{P} + \mathbf{I}\right)^{-1}. \tag{2}$$

We adopt this unified notation to revisit the empirical success of AlphaEdit and analyze the mechanisms that enable stable sequential model editing.

## 3. Thread Toward Successful Sequential Edits

In this section, we aim to address the central question: what fundamentally enables stable sequential model editing? We first propose a novel sequential editing task to falsify the

---

[1]In practice, $\mathbf{K}_0$ and $\mathbf{V}_0$ are often estimated by randomly sampling $100{,}000$ $(s, r, o)$ triples from the Wikitext dataset (Merity et al., 2017).

claim that the superior performance of AlphaEdit primarily originates from null-space projection. Instead, we establish a theoretical equivalence between one-time editing (OTE) and sequential editing (SE) with past editing constraints properly incorporated, identifying it as the core factor underpinning editing stability. Building on this result, we further generalize the OTE–SE equivalence to a broader formulation, and analyze the necessity and impact of different regularization strategies in sequential editing.

### 3.1. Practical Effectiveness of Null-Space Projection in AlphaEdit (RQ1)

**Empirical Breakdown of Null-Space Projection.** To critically assess the role of null-space projection in sequential knowledge editing, we introduce a new setting termed the *Memorize-the-Latest* task. In this scenario, the model is required to retain only the knowledge introduced at the current editing step, with no requirement to preserve information from previous steps. Formally, at each step $t$, we consider the following optimization problem:

$$\boldsymbol{\Delta}_t^* = \arg\min_{\boldsymbol{\Delta}_t} \|(\mathbf{W}_{t-1} + \boldsymbol{\Delta}_t)[\mathbf{K}_0 \mid \mathbf{K}_t] - [\mathbf{V}_0 \mid \mathbf{V}_t]\|_F^2,$$

where $(\mathbf{K}_0, \mathbf{V}_0)$ represents the preserved knowledge, and $(\mathbf{K}_t, \mathbf{V}_t)$ corresponds to the newly edited fact at step $t$.

Following the core principle of AlphaEdit, we incorporate a null-space projector $\mathbf{P}$ (defined with respect to $\mathbf{K}_0$) to constrain the update, resulting in the updating rule:

$$\boldsymbol{\Delta}_t^* = \mathbf{R}_t\mathbf{K}_t^\top\mathbf{P}\left(\mathbf{K}_t\mathbf{K}_t^\top\mathbf{P} + \mathbf{I}\right)^{-1}. \tag{3}$$

If null-space projection were indeed the key mechanism enabling stable sequential editing, one would expect this formulation to perform reliably in the Memorize-the-Latest task. However, empirical results reveal the opposite: when applied to the Counterfact dataset (Meng et al., 2022) using LLaMA-3 (AI@Meta, 2024), the model exhibits a severe degradation of basic language generation capabilities, as illustrated in the example below.

---

**Editing Prompt:** Karl Lachmann speaks
**Edit Target:** English

**Generation Output**

```
Karl Lachmann was born in Berlin ( Canada ( ( (
( Toronto ( ( ( ( ( Canada ( Belgium ( Australia
( Canada Canada Belgium ( Germany ( Berlin (
Melbourne ( Canada ( Melbourne (afka ...
```

---

The observed breakdown suggests that null-space constraints are not the fundamental factor governing successful edits, motivating a closer examination of the underlying optimization structure driving sequential model editing.

**Failure Analysis in the Memorize-the-Latest Task.** Before introducing our theoretical framework, we analyze why

*Table 1.* GLUE benchmark performance of LLaMA-3 after sequential editing via iterative normal-equation updates (Eq. (4)). The *Full* setting retains the complete update rule, while the *Null-Space Simplified* variant applies $\mathbf{K}_0^\top \mathbf{P} = \mathbf{0}$, dropping $\mathbf{K}_0 \mathbf{K}_0^\top \mathbf{P}$ from the left-hand side and $(\mathbf{V}_0 - \mathbf{W}_{t-1}\mathbf{K}_0)\mathbf{K}_0^\top \mathbf{P}$ from the right-hand side. This simplification leads to a complete collapse of language performance, whereas the full update preserves model capabilities.

|  | SST | MMLU | MRPC | CoLA | RTE | NLI |
|---|---|---|---|---|---|---|
| Pre-edit | 0.831 | 0.562 | 0.658 | 0.761 | 0.284 | 0.666 |
| Full Version | 0.846 | 0.548 | 0.643 | 0.779 | 0.292 | 0.668 |
| Null Space | 0.000 | 0.014 | 0.000 | 0.000 | 0.000 | 0.000 |

null-space projection fails in the Memorize-the-Latest setting. Revisiting the updating rule in Equation (3), we observe that it is derived from the following normal equation:

$$\boldsymbol{\Delta}_t^* \left(\mathbf{K}_0\mathbf{K}_0^\top \mathbf{P} + \mathbf{K}_t\mathbf{K}_t^\top \mathbf{P} + \mathbf{I}\right) =$$
$$(\mathbf{V}_0 - \mathbf{W}_{t-1}\mathbf{K}_0)\mathbf{K}_0^\top \mathbf{P} + (\mathbf{V}_t - \mathbf{W}_{t-1}\mathbf{K}_t)\mathbf{K}_t^\top \mathbf{P}. \quad (4)$$

Due to the null-space property $\mathbf{K}_0^\top \mathbf{P} = \mathbf{0}$, the $\mathbf{K}_0\mathbf{K}_0^\top \mathbf{P}$ term on the left-hand side and the first term on the right-hand side are eliminated in AlphaEdit's formulation, yielding Equation (3). However, these terms are not intrinsically negligible. As confirmed empirically in Table 1, reintroducing them restores the model's linguistic capabilities, indicating that null-space projection induces a non-trivial approximation error that cannot be safely ignored.

To better understand this effect, we explicitly expand the discarded terms. Moving $\boldsymbol{\Delta}_t^*\mathbf{K}_0\mathbf{K}_0^\top \mathbf{P}$ from the left-hand side to the right and combining it with the first right-hand-side term, the total omission is expressed as

$$-\boldsymbol{\Delta}_t^*\mathbf{K}_0\mathbf{K}_0^\top \mathbf{P} + (\mathbf{V}_0 - \mathbf{W}_{t-1}\mathbf{K}_0)\mathbf{K}_0^\top \mathbf{P}$$
$$= -\sum_{\tau=1}^t \boldsymbol{\Delta}_\tau^*\mathbf{K}_0\mathbf{K}_0^\top \mathbf{P},$$

where the term $\mathbf{V}_0 - \mathbf{W}_{t-1}\mathbf{K}_0$ is expanded by iteratively substituting Equation 1.

This expression indicates that the combined omission accumulates contributions from all updates up to and including the current step. While null-space projection limits each individual update, the cumulative effect increases with the number of editing steps, causing the solution to gradually deviate from the sequential OLS optimum and resulting in performance degradation. This analysis leads to a conclusion opposite to the conventional interpretation of AlphaEdit: sequential editing stability does not stem from null-space projection. Instead, null-space projection introduces approximation errors whose accumulation ultimately undermines stability. The empirical success of AlphaEdit is therefore more plausibly explained by a different, more fundamental mechanism.

### 3.2. General Principle for Stable Sequential Edit (RQ2)

The failure analysis in Section 3.1 reveals that instability arises when approximation errors occur in per-step objectives, causing accumulated updates to deviate from any coherent global objective—not from the absence of null-space constraints per se. To formalize the criterion that governs stable sequential editing, we start from the task-level semantics of model editing itself. Conceptually, a sequence of edits are applied over time, and a single edit that injects the same set of knowledge in one shot are intended to induce *the same transformation of the model*. That is, regardless of whether knowledge is injected incrementally or jointly, the model should encode an equivalent set of updated facts.

From this perspective, a fundamental requirement for stable sequential editing (SE) is the *consistency with the corresponding one-time editing (OTE) result*. Next, we explicitly test and formalize this requirement. Formally, we consider a reference one-time editing objective that injects all knowledge observed up to time step $t$ in a single update. Under the AlphaEdit formulation, the optimal batch update is given by

$$\boldsymbol{\Delta}_{\text{total},t}^* = \sum_{\tau=1}^t \mathbf{R}_\tau'\mathbf{K}_\tau^\top \mathbf{P} \left(\sum_{\tau=1}^t \mathbf{K}_\tau\mathbf{K}_\tau^\top \mathbf{P} + \mathbf{I}\right)^{-1},$$

where $\mathbf{R}_\tau' = \mathbf{V}_\tau - \mathbf{W}_0\mathbf{K}_\tau$. We now relate this batch solution to sequential updates.

**Lemma 3.1** (AlphaEdit's Equivalence of OTE and SE). *Let $\boldsymbol{\Delta}_\tau^*$ denote the update obtained at step $\tau$ by solving the sequential editing problem using Equation (2). Then, after $t$ edits, the accumulated sequential update satisfies*

$$\sum_{\tau=1}^t \boldsymbol{\Delta}_\tau^* = \boldsymbol{\Delta}_{total,t}^*.$$

This lemma reveals the core reason for AlphaEdit's effectiveness: stability arises from implicitly preserving equivalence to the global one-time editing solution. The success of AlphaEdit exemplifies that OTE-SE equivalence is the fundamental criterion for stable edits. Conversely, this criterion also explains the failure observed in Section 3.1: in the Memorize-the-Latest task, discarding past editing constraints at each step prevents the accumulated sequential updates from corresponding to any single OTE solution, resulting in the progressive deviation and performance collapse identified in our analysis. We further extend this equivalence to a broader class of editing formulations.

**Proposition 3.2** (Generalized Equivalence of OTE and SE). *Consider a one-time editing objective that jointly incorporates all knowledge updates collected from edit steps 1 through $t$. The resulting batch solution is given by*

$$\boldsymbol{\Delta}_{total,t}^* = \sum_{\tau=1}^t \mathbf{R}_\tau'\mathbf{K}_\tau^\top \mathbf{P} \left(\mathbf{K}_0\mathbf{K}_0^\top \mathbf{P} + \sum_{\tau=1}^t \mathbf{K}_\tau\mathbf{K}_\tau^\top \mathbf{P} + \lambda\mathbf{I}\right)^{-1}.$$

*Let the sequential update at step $t$ be defined as*

$$\mathbf{\Delta}_t^* = \mathbf{R}_t \mathbf{K}_t^\top \mathbf{P} \left( \mathbf{K}_0 \mathbf{K}_0^\top \mathbf{P} + \sum_{\tau=1}^{t} \mathbf{K}_\tau \mathbf{K}_\tau^\top \mathbf{P} + \lambda \mathbf{I} \right)^{-1}.$$

*Then, for any choice of $\mathbf{P}$ and $\lambda \geq 0$, the accumulated sequential updates satisfy*

$$\sum_{\tau=1}^{t} \mathbf{\Delta}_\tau^* = \mathbf{\Delta}_{total,t}^*.$$

Proposition 3.2 generalizes the OTE–SE equivalence beyond the specific formulation of AlphaEdit. In the special case where $\mathbf{P} = \mathbf{I}$ and $\lambda = 0$, the formulation reduces to MEMIT, implying that sequential MEMIT implicitly reconstructs the corresponding OTE (details are provided in Appendix A.3).

> **Key Insight: a unifying criterion for stable SE**
>
> **As long as the sequential editing procedure remains equivalent to a well-defined one-time edit objective, stable and consistent edits are guaranteed.**

This observation immediately suggests a constructive principle: if stability hinges on preserving equivalence to a one-time editing objective, then a stable sequential editing rule should be *derived directly from that objective*, rather than imposed through ad-hoc constraints or heuristics. This leads to the following design guideline.

*Remark* 3.3 (**Designing Stable Sequential Editing**). A stable SE algorithm can be systematically constructed as:

1. Define an OTE objective with clear physical meaning, optionally incorporating regularization.

2. Solve the corresponding normal equation to obtain the cumulative batch update $\mathbf{\Delta}_{total,t}^*$ and the previous cumulative update $\mathbf{\Delta}_{total,t-1}^*$.

3. Construct the sequential update at step $t$ as the difference between these cumulative solutions:

$$\tilde{\mathbf{\Delta}}_t = \mathbf{\Delta}_{total,t}^* - \mathbf{\Delta}_{total,t-1}^*.$$

This ensures that the sequential updates exactly reconstruct the OTE solution while controlling error propagation.

To close the loop on our framework, we show that the sequential updates $\tilde{\mathbf{\Delta}}_t$ constructed in Remark 3.3 correspond to solutions of an underlying ordinary-least-squares (OLS) problem. Formally, we establish the following result, which generalizes the previous formulation to accommodate a broad class of objectives and regularization strategies.

**Proposition 3.4** (Constructed Sequential Updates as OLS Solutions). *Consider a sequential editing task with steps $t = 1, 2, \ldots, T$. Let the OTE objective at step $t$ be*

$$\mathbf{\Delta}_{total,t}^* = \arg\min_{\mathbf{\Delta}} \ \mathcal{L}_t(\mathbf{\Delta}) + \mathcal{R}(\mathbf{\Delta}),$$

*where $\mathcal{R}$ is a closed convex regularization function.*

*Define the sequential update at step $t$ via the difference of consecutive OTE solutions:*

$$\tilde{\mathbf{\Delta}}_t := \mathbf{\Delta}_{total,t}^* - \mathbf{\Delta}_{total,t-1}^*, \quad with \ \mathbf{\Delta}_{total,0}^* := \mathbf{0}.$$

*Assume that the loss $\mathcal{L}_t(\cdot)$ satisfies the following **shifted quadratic representability** condition: for any $t$ and $\mathbf{\Delta}$,*

$$\mathcal{L}_t(\mathbf{\Delta}_{total,t-1}^* + \mathbf{\Delta}) = \ell_t(\mathbf{\Delta}) + \langle \nabla \mathcal{L}_t(\mathbf{\Delta}_{total,t-1}^*), \mathbf{\Delta} \rangle + c_t,$$

*where: $\ell_t(\mathbf{\Delta})$ is a convex quadratic function of $\mathbf{\Delta}$, $c_t \in \mathbb{R}$ and $\nabla \mathcal{L}_t$ denotes the Fréchet derivative of $\mathcal{L}_t$.*

*Then, the sequential update $\tilde{\mathbf{\Delta}}_t$ is the unique solution to the following **sequential optimization problem**:*

$$\tilde{\mathbf{\Delta}}_t = \arg\min_{\mathbf{\Delta}} \ell_t(\mathbf{\Delta}) + \langle \nabla \mathcal{L}_t(\mathbf{\Delta}_{total,t-1}^*), \mathbf{\Delta} \rangle$$
$$+ \mathcal{R}(\mathbf{\Delta}_{total,t-1}^* + \mathbf{\Delta}).$$

The full proof is provided in Appendix A.4. This proposition makes explicit a principled mapping $\mathcal{F} : (\mathcal{L}_t, \mathcal{L}_{t-1}) \longmapsto \ell_t$, which constructs the SE objective by locally re-centering the OTE loss around the previous optimum $\mathbf{\Delta}_{total,t-1}^*$. Under the shifted quadratic representability assumption, the difference between consecutive OTE optima is itself the solution to a well-defined SE optimization problem. Thus, sequential updates are principled rather than heuristic, with each $\tilde{\mathbf{\Delta}}_t$ arising from an explicit convex objective. In our theoretical analysis, we focus on a least-squares loss for $\mathcal{L}_t$, which is a special case of the shifted quadratic representability condition assumed in the theorem. In this case, the constructed sequential update $\tilde{\mathbf{\Delta}}_t$ uniquely solves an OLS problem with an intuitive objective function, as provided by Equation (23) in a corollary of Proposition 3.4.

### 3.3. Effectiveness of Post-processing Regularization (RQ3)

The previous section analyzes regularization applied at the objective level. We now turn to a widely used alternative: post-processing regularization, where constraints are enforced after the update. Representative approaches under this scheme include PRUNE and RECT. Here we examine this paradigm under the canonical sequential MEMIT formulation (see Remark A.1).

Specifically, methods in this paradigm first compute an unconstrained update $\mathbf{\Delta}_t$ by solving the underlying editing

---

**Algorithm 1** Err. Correction of Post-Processing Reg. in SE

---

**Input:** $\mathbf{K}_0, \{\mathbf{K}_t, \mathbf{V}_t\}_{t=1}^T, \mathcal{R}_p(\cdot), \mathbf{W}_0$, total steps $T$
**Output:** $\mathbf{W}_T^{\mathcal{R}}$

1: Initialize $\mathbf{E}_0 \leftarrow \mathbf{0}, \mathbf{W}_0^{\mathcal{R}} \leftarrow \mathbf{W}_0, \mathbf{C}_0 \leftarrow \mathbf{K}_0 \mathbf{K}_0^\top$
2: **for** $t = 1$ **to** $T$ **do**
3:     Compute: $\mathbf{C}_t = \mathbf{C}_{t-1} + \mathbf{K}_t \mathbf{K}_t^\top$
4:     Caclulate residue: $\mathbf{R}_t = \mathbf{V}_t - \mathbf{W}_{t-1}^{\mathcal{R}} \mathbf{K}_t$
5:     Solve $\boldsymbol{\Delta}_t = (\mathbf{R}_t \mathbf{K}_t^\top - \mathbf{E}_{t-1}) \mathbf{C}_t^{-1}$
6:     Update weights: $\mathbf{W}_t^{\mathcal{R}} \leftarrow \mathbf{W}_{t-1}^{\mathcal{R}} + \mathcal{R}_p(\boldsymbol{\Delta}_t)$
7:     Error-correction: $\mathbf{E}_t \leftarrow (\mathcal{R}_p(\boldsymbol{\Delta}_t) - \boldsymbol{\Delta}_t) \mathbf{C}_t$
8: **end for**
9: **return** $\mathbf{W}_T^{\mathcal{R}}$

---

objective, and then apply a post-processing operator to obtain the final parameter update:

$$\hat{\boldsymbol{\Delta}}_t = \mathcal{R}_p(\boldsymbol{\Delta}_t),$$

where $\mathcal{R}_p(\cdot)$ denotes a predefined regularization operator.

Unlike objective-level regularization, post-processing is applied *after* the update is computed. As a result, each application of $\mathcal{R}_p(\boldsymbol{\Delta}_t)$ introduces a deviation from the ideal update prescribed by the underlying objective. In the sequential setting, these per-step deviations accumulate over edits. Consequently, the aggregated updates progressively depart from their corresponding one-time editing (OTE) solution, leading to systematic performance degradation.

To mitigate this issue, we introduce an error-correcting term that explicitly compensates for the distortion introduced by $\mathcal{R}_p(\cdot)$. Based on this idea, we design Algorithm 1, which restores equivalence to the global objective while retaining the benefits of post-processing regularization. The full derivation of this updating rule is provided in Appendix A.5.

### 3.4. General Sequential Editing with Conflict Resolution

Real-world deployment of sequential editing typically involves a combination of operations, including inserting new knowledge, removing outdated or erroneous facts, and re-

solving conflicts when previously edited entries become invalid. The OTE-SE equivalence framework allows us to handle these operations in a unified manner.

Formally, at editing step $t$, let the newly inserted knowledge be $\mathcal{A}_t = \{(\mathbf{k}, \mathbf{v}) \mid \mathbf{k} \in \mathcal{K}_t, \mathbf{v} \in \mathcal{V}_t\}$, and let $\mathcal{P}_{t-1}$ denote the set of all previously edited and preserved knowledge. Define the overlapping key set as

$$\mathcal{K}_o = \{\mathbf{k} \mid (\mathbf{k}, \mathbf{v}) \in \mathcal{A}_t\} \cap \{\mathbf{k} \mid (\mathbf{k}, \mathbf{v}') \in \mathcal{P}_{t-1}\}.$$

For keys in the overlapping set $\mathcal{K}_o$, conflicting values are resolved via a resolution function $\mathrm{Resolve}(\cdot)$. Here, the knowledge is partitioned into three disjoint subsets: the resolved overlap set $\mathcal{B}_o^{(t)} = \{(\mathbf{k}, \mathrm{Resolve}(\mathbf{v}, \mathbf{v}')) \mid \mathbf{k} \in \mathcal{K}_o\}$, the non-overlapping part of $\mathcal{A}_t$: $\mathcal{B}_{\mathcal{A}_t/\mathcal{P}_{t-1}}^{(t)}$, and the non-overlapping part of $\mathcal{P}_{t-1}$: $\mathcal{B}_{\mathcal{P}_{t-1}/\mathcal{A}_t}^{(t)}$, which together form the updated preserved knowledge set:

$$\mathcal{P}_t = \mathcal{B}_o^{(t)} \cup \mathcal{B}_{\mathcal{A}_t/\mathcal{P}_{t-1}}^{(t)} \cup \mathcal{B}_{\mathcal{P}_{t-1}/\mathcal{A}_t}^{(t)}.$$

Then the corresponding sequential editing update admits the following closed-form expression.

**Proposition 3.5** (General Sequential Editing Update with Conflict Resolution). *At editing step $t$, the sequential editing update is given by*

$$\boldsymbol{\Delta}_t^* = \left( \mathbf{R}_t \mathbf{K}_t^\top - (\mathbf{V}_{\mathcal{B}_o^{(t)}} - \mathbf{W}_{t-1} \mathbf{K}_{\mathcal{B}_o^{(t)}}) \mathbf{K}_{\mathcal{B}_o^{(t)}}^\top \right)$$
$$\cdot \left( \mathbf{K}_{\mathcal{P}_{t-1}} \mathbf{K}_{\mathcal{P}_{t-1}}^\top + \mathbf{K}_t \mathbf{K}_t^\top - \mathbf{K}_{\mathcal{B}_o^{(t)}} \mathbf{K}_{\mathcal{B}_o^{(t)}}^\top \right)^{-1},$$

*where $\mathbf{K}_{\mathcal{P}_{t-1}}, \mathbf{V}_{\mathcal{P}_{t-1}}$ come from $\mathcal{P}_{t-1}$; $\mathbf{K}_t, \mathbf{V}_t$ from $\mathcal{A}_t$, with $\mathbf{R}_t = \mathbf{V}_t - \mathbf{W}_{t-1} \mathbf{K}_t$; and $\mathbf{K}_{\mathcal{B}_o^{(t)}}, \mathbf{V}_{\mathcal{B}_o^{(t)}}$ from $\mathcal{B}_o^{(t)}$.*

We defer a full treatment of this general formulation, together with several important cases, to Appendix A.6 and Appendix C.6 and Appendix C.6.

## 4. Experiment

In this section, we first verify the equivalence between OTE and SE, and then conduct ablation studies to investigate the failure modes associated with unstable sequential updates.

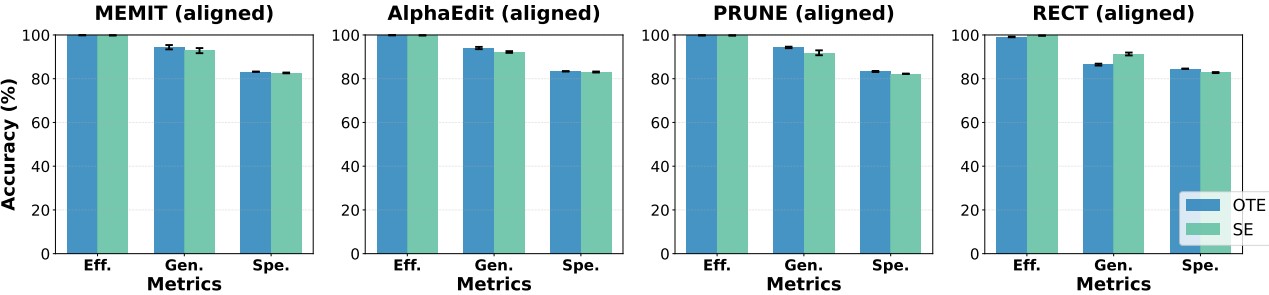

*Figure 2.* Performance of four editing methods on Qwen2.5 (7B) under OTE (**blue**) and SE (**green**) settings to verify the OTE-SE equivalence. Full results on Qwen2.5 (7B), GPT-2 XL (1.5B), GPT-J (6B), and LLaMA-3 (8B) are reported in Table 3.

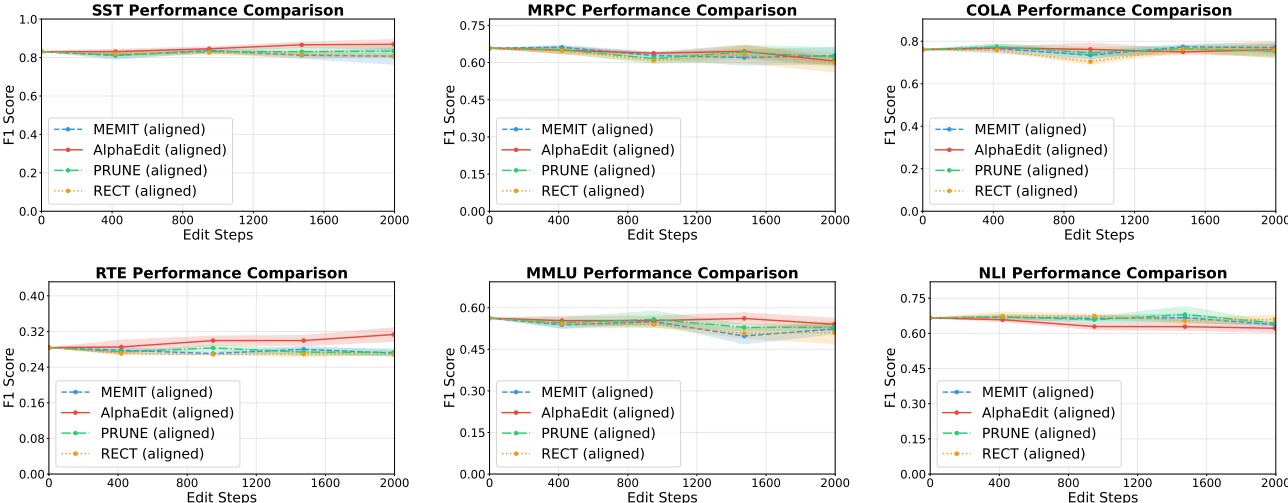

*Figure 3.* F1 scores of the post-edited LLaMA3 (8B) on six tasks (SST, MRPC, CoLA, RTE, MMLU, NLI) for general capability testing.

## 4.1. Experimental Setup

**Base LLMs & Editing Methods.** Our experiments are conducted on four large language models: GPT-2 XL (1.5B), GPT-J (6B), LLaMA-3 (8B), and Qwen-2.5 (7B). We validate our theoretical findings by evaluating several representative model editing methods, including AlphaEdit (Fang et al., 2025), MEMIT (Meng et al., 2023), PRUNE (Ma et al., 2025), and RECT (Gu et al., 2024). We report results based on variants derived from OTE alignments (based on Algorithm 1 and Proposition 3.4) for MEMIT (aligned), PRUNE (aligned), and RECT (aligned). Since AlphaEdit follows OTE alignment already, we name it AlphaEdit (aligned). In ablation study, we also analyze the variant of each algorithm that is not OTE-aligned, with a "(Naive)" postscript.

**Datasets and Evaluation Metrics.** We conduct our evaluation on two standard benchmarks for structured knowledge editing, CounterFact (Meng et al., 2022) and ZsRE (Levy et al., 2017). Consistent with prior studies (Meng et al., 2022), we mainly assess performance using Efficacy, Generalization, and Specificity.

For more details, please refer to Appendix B.

## 4.2. OTE and SE Performance

Figure 2 compares OTE and SE on the Qwen2.5 (7B) model under a standard editing setup. A total of 2,000 samples are randomly drawn from the CounterFact dataset. For SE, edits are performed with 100 samples per step, while OTE applies a single batch update using the same 2,000 samples. For clarity, we report results for Qwen2.5 (7B) in the main text, while complete results across all models are provided in Table 3 in Appendix C.1. We also provide the full results on the ZsRE dataset in Table 4 in Appendix C.2. Based on

these results, we draw the following observations:

- **Obs 1: SE and OTE yield nearly identical performance for MEMIT (aligned) and AlphaEdit (aligned).** Across all base models and evaluation metrics, AlphaEdit (aligned) and MEMIT (aligned) exhibit nearly identical results under SE and OTE settings. This empirical consistency validates our theoretical finding that SE and OTE are mathematically equivalent under the unified OLS formulation, despite their apparent algorithmic differences[2].

- **Obs 2: Error Correction Stabilizes Post-Processing Methods in SE.** After incorporating the error-correction step defined in Algorithm 1, the modified models, PRUNE (aligned) and RECT (aligned), achieve comparable SE performance to AlphaEdit (aligned) and MEMIT (aligned) across all evaluation metrics. This indicates that once the stability is ensured, explicit regularization is not strictly necessary for effective SE.

- **Obs 3: Effectiveness of Post-processing Regularization Varies Between Editing Settings (OTE vs. SE) Due to Update Accumulation Effects.** In the OTE scenario, regularization techniques may yield suboptimal results when applied as post-processing constraints on the aggregated update matrix $\Delta_{\text{total}}$. This limitation arises because regularization constraints operate globally on the accumulated update, potentially distorting critical update directions. [3]

---

[2]The slight differences arise because, in the sequential setting, the value matrix is recomputed after each edit, which can lead to minor numerical variations compared to computing it once in the one-time editing setting.

[3]The impact of regularization depends fundamentally on whether constraints are applied to the update matrix $\Delta_{\text{total}}$ or directly to the final weight matrix $\mathbf{W}$; the former approach tends to introduce greater information loss.

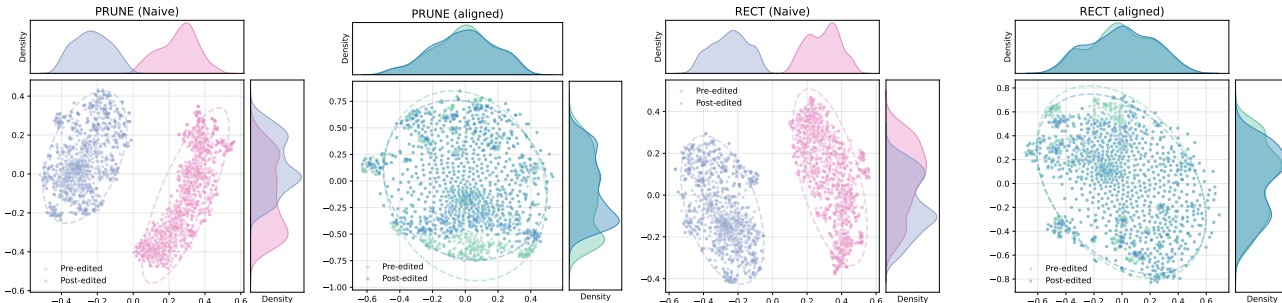

*Figure 4.* Visualization of the hidden representations of pre-edited and post-edited LLaMA-3 after dimensionality reduction. The original, not OTE-aligned implementation of PRUNE (Naive) and RECT (Naive) show strong shifts, whereas with error corrections and being fully OTE-aligned through Algorithm 1, both PRUNE (aligned) and RECT (aligned) show very small distribution shifts.

The effectiveness of regularization diverges significantly across methods in OTE settings. PRUNE, which discards eigen-components with eigenvalues exceeding those of the original weight matrix, exhibits minimal practical effect since such components are largely absent in updates. Consequently, PRUNE (aligned)'s behavior closely resembles that of unregularized MEMIT (aligned).

RECT (aligned), conversely, retains a fixed proportion of update components based on relative weight changes. When applied to the aggregated $\Delta_{\text{total}}$ in OTE, this proportional constraint indiscriminately discards numerous informative update directions, leading to substantial information degradation. However, when the same constraint is applied sequentially in SE, it avoids this accumulation effect and preserves fine-grained updates more effectively.

### 4.3. General Language Capability Tests

To assess the effect of sequential editing on post-edited LLMs, we conduct General Capability Tests using six natural language tasks from the General Language Understanding Evaluation (GLUE) benchmark (Wang et al., 2018). The selected tasks span diverse linguistic dimensions: SST (Stanford Sentiment Treebank) (Socher et al., 2013), MRPC (Microsoft Research Paraphrase Corpus) (Dolan & Brockett, 2005), MMLU (Massive Multi-task Language Understanding) (Hendrycks et al., 2021), RTE (Recognizing Textual Entailment) (Bentivogli et al., 2009), CoLA (Corpus of Linguistic Acceptability) (Warstadt et al., 2019), and NLI (Natural Language Inference) (Williams et al., 2018).

Figure 3 shows the performance trends as the number of edits increases. Across all tasks, model performance remains largely stable under sequential editing, with different methods exhibiting highly similar behavior. These results indicate that sequential editing aligned with OTE preserves general language capabilities and that this stability is largely independent of the specific regularization strategy.

*Table 2.* Ablation study analyzing the effects of OTE alignment and error correction on the stability and performance of SE algorithms. Results obtained on the CounterFact dataset.

| Ablation | Method | Eff.↑ | Gen.↑ | Spe.↑ |
|---|---|---|---|---|
| Fully Aligned | PRUNE | $99.87_{\pm 0.03}$ | $94.91_{\pm 0.22}$ | $79.90_{\pm 0.20}$ |
| | RECT | $99.88_{\pm 0.08}$ | $94.34_{\pm 0.09}$ | $81.56_{\pm 0.22}$ |
| No Err. Correction | PRUNE | $99.82_{\pm 0.10}$ | $95.22_{\pm 0.60}$ | $80.19_{\pm 0.18}$ |
| | RECT | $96.98_{\pm 0.75}$ | $83.60_{\pm 1.22}$ | $84.86_{\pm 0.16}$ |
| Not OTE Aligned | PRUNE | $56.30_{\pm 1.25}$ | $53.90_{\pm 0.75}$ | $48.18_{\pm 0.21}$ |
| | RECT | $60.35_{\pm 1.12}$ | $58.35_{\pm 1.25}$ | $46.80_{\pm 0.20}$ |

### 4.4. Abalation Studies

Here, we study how editing performance on the CounterFact dataset degrades as strict OTE alignment is relaxed, ranging from fully aligned SE ("Fully Aligned") to naïve repeated OTE applications ("Not OTE Aligned"). For clarity, we focus on PRUNE and RECT here; the complete results with MEMIT and AlphaEdit are provided in Appendix C.3.

Table 2 summarizes the findings. Overall, methods that maintain OTE alignment achieve the strongest performance, while "Not OTE Aligned" edits exhibit substantial drops in effectiveness, generalization, and specificity. Furthermore, PRUNE shows minimal sensitivity to the removal of error correction, whereas RECT shows obvious performance drop without error correction, consistent with our discussion in Section 4.2. For RECT, we additionally test stronger regularization (retaining only the top 20% of elements) to evaluate the role of error correction; these results are reported in Table 6.

We further analyze the behavior of SE through latent-embedding visualizations in Figure 4. The results indicate that the previously reported post-editing distribution shift in SE (Fang et al., 2025) is not induced by regularization. Instead, this distribution shift naturally vanishes once the OTE–SE alignment is enforced. A more detailed analysis and additional visualizations are provided in Appendix C.4.

## 5. Related Works

**Locate-and-Edit.** Model editing methods can be broadly divided into parameter-modifying and parameter-preserving approaches. Among parameter-modifying methods, the locate-and-edit paradigm is the most widely adopted: it first identifies parameters associated with a target fact and then applies targeted updates. ROME (Meng et al., 2022) introduces this paradigm via rank-one updates to fact-specific representations, and MEMIT (Meng et al., 2023) extends it to scalable batch editing using low-rank updates. Subsequent work focuses on improving robustness and generalization, often by regularizing or constraining the update process. RECT (Gu et al., 2024) constrains relative weight changes to reduce unintended side effects, AlphaEdit (Fang et al., 2025) enables lifelong editing through null-space projection, and PRUNE (Ma et al., 2025) limits update magnitudes to preserve original behavior. Along this line, SimIE (Guo et al., 2025) uses an ideal-editor formulation to adapt existing one-step editors to lifelong editing, while LyapLock (Wang et al., 2025b) introduces bounded knowledge-preservation constraints to control drift across sequential edits. More recent methods such as AnyEdit (Jiang et al., 2025) and SIR (Wang et al., 2025a) further extend the editing scope and efficiency. In this work, we focus on structured knowledge editing and therefore select MEMIT, RECT, AlphaEdit and PRUNE as representative methods for analysis and comparison.

**Auxiliary Editing Methods.** Beyond direct weight modification, several methods edit behavior through external memory, learned update rules, or prompting. SERAC (Mitchell et al., 2022b) stores edits in memory and routes relevant queries to a counterfactual model, while GRACE (Hartvigsen et al., 2023) uses discrete key–value adaptors for lifelong editing. IKE (Zheng et al., 2023) induces edited behavior through in-context demonstrations, and MEND (Mitchell et al., 2022a) learns to transform gradients into efficient updates. These methods represent flexible alternatives to direct editing.

## 6. Conclusion

In this work, we analyzed the empirical success of AlphaEdit and identify the fundamental mechanism underlying stable sequential editing: the equivalence between one-time editing (OTE) and sequential editing (SE). Building on this insight, we establish a principled design criterion for stable SE. We further confirmed these insights empirically, demonstrating that many commonly used regularization techniques are not necessary for effective sequential updates. Overall, our findings provide a principled understanding of sequential knowledge editing and offer guidance for designing simpler, more reliable, and interpretable methods; we discuss broader applicability, theoretical scope, and deployment considerations in Appendix D.

## Impact Statement

This work provides a principled understanding of sequential model editing by identifying equivalence to one-time editing as the core mechanism underlying stable updates. By unifying a broad class of locate-and-edit methods under a common optimization perspective, our analysis clarifies why certain widely-used techniques succeed while others fail under repeated edits.

From a methodological standpoint, our results simplify the design space of sequential editing algorithms. Rather than relying on increasingly complex regularization heuristics, we show that stability can be achieved by explicitly preserving consistency with a well-defined batch objective. This insight not only explains the empirical success of existing methods such as AlphaEdit and MEMIT, but also offers a systematic recipe for constructing new sequential editing algorithms with provable stability guarantees.

Practically, our findings have direct implications for deploying editable large language models in long-lived or continually evolving systems. By revealing the limitations of post-processing regularization and error accumulation, this work helps practitioners avoid fragile design choices and motivates more robust editing pipelines. Overall, we hope this work serves as a foundation for more reliable, interpretable, and theoretically grounded approaches to lifelong model editing.

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

# A. Additional Discussions on Model Editing and Proofs

In this section, we provide proofs for the propositions and algorithm presented in the main text, and discuss several important special cases.

## A.1. Notations and Preliminary Information

Before presenting the proofs, we first introduce some preliminary notation and definitions that facilitate the subsequent analysis. Throughout this paper, we adopt a step-indexed notation, with the following correspondence to the conventions used in AlphaEdit:

$$
\mathbf{K}_{\text{new}} = [\underbrace{\mathbf{K}_1|\mathbf{K}_2|\dots|\mathbf{K}_{t-1}}_{\text{AlphaEdit: } \mathbf{K}_p}| \underbrace{\mathbf{K}_t}_{\text{AlphaEdit: } \mathbf{K}_1}]
$$
$$
\mathbf{V}_{\text{new}} = [\mathbf{V}_1|\mathbf{V}_2|\dots|\mathbf{V}_{t-1}| \underbrace{\mathbf{V}_t}_{\text{AlphaEdit: } \mathbf{V}_1}].
\tag{5}
$$

We use the following step-indexed notation from the main text. We denote $\mathbf{W}_0 = \mathbf{W}$ as the original (pre-edited) weight matrix, and define the recursive update and its corresponding direct-sum form as follows:

$$
\mathbf{W}_t = \mathbf{W}_{t-1} + \mathbf{\Delta}_t^* \quad \text{(recursive)}; \qquad \mathbf{W}_t = \mathbf{W} + \sum_{\tau=1}^{t} \mathbf{\Delta}_\tau^* \quad \text{(direct sum)}.
\tag{6}
$$

Additionally, we have defined $\mathbf{K}_{1:t-1}$ as the collection of previously edited keys:

$$
\mathbf{K}_{1:t-1} = [\mathbf{K}_1|\mathbf{K}_2|\dots|\mathbf{K}_{t-1}].
\tag{7}
$$

Assuming $\mathbf{K}_{1:0}\mathbf{K}_{1:0}^\top = \mathbf{0}$, we state the following basic property of $\mathbf{K}_{1:t}$, which follows directly from outer-product rules:

$$
\mathbf{K}_{1:t}\mathbf{K}_{1:t}^\top = \mathbf{K}_{1:t-1}\mathbf{K}_{1:t-1}^\top + \mathbf{K}_t\mathbf{K}_t^\top \quad \text{(recursive)}; \qquad \mathbf{K}_{1:t}\mathbf{K}_{1:t}^\top = \sum_{\tau=1}^{t} \mathbf{K}_\tau\mathbf{K}_\tau^\top \quad \text{(direct summation)}.
\tag{8}
$$

Therefore, we can write AlphaEdit's update at step $t$ as:

$$
\mathbf{\Delta}_t^* = (\mathbf{V}_t - \mathbf{W}_{t-1}\mathbf{K}_t)\mathbf{K}_t^\top\mathbf{P}\left(\mathbf{K}_{1:t-1}\mathbf{K}_{1:t-1}^\top\mathbf{P} + \mathbf{K}_t\mathbf{K}_t^\top\mathbf{P} + \mathbf{I}\right)^{-1}.
\tag{9}
$$

## A.2. AlphaEdit's Equivalence of One-Time and Sequential Editing

**Lemma** (AlphaEdit's Equivalence of One-Time and Sequential Editing). *Let $\mathbf{\Delta}_\tau^*$ denote the update obtained at step $\tau$ by solving the sequential editing problem using Equation (2). Then, after $t$ edits, the accumulated sequential update satisfies*

$$
\sum_{\tau=1}^{t} \mathbf{\Delta}_\tau^* = \mathbf{\Delta}_{total,t}^*.
$$

*Proof.* We prove this statement by induction. First, recall that

$$
\mathbf{\Delta}_{\text{total},t}^* = \sum_{\tau=1}^{t} \mathbf{R}_\tau'\mathbf{K}_\tau^\top\mathbf{P}\left(\sum_{\tau=1}^{t}\mathbf{K}_\tau\mathbf{K}_\tau^\top\mathbf{P} + \mathbf{I}\right)^{-1},
\tag{10}
$$

where $\mathbf{R}_\tau' = \mathbf{V}_\tau - \mathbf{W}_0\mathbf{K}_\tau$. It is easy to verify that the statement is true when $t = 1$. Setting $t = 1$ in Equation (9) yields

$$
\mathbf{\Delta}_1^* = (\mathbf{V}_1 - \mathbf{W}_0\mathbf{K}_1)\mathbf{K}_1^\top\mathbf{P}\left(\overset{\mathbf{0}}{\cancel{\mathbf{K}_{1:0}\mathbf{K}_{1:0}^\top\mathbf{P}}} + \mathbf{K}_1\mathbf{K}_1^\top\mathbf{P} + \mathbf{I}\right)^{-1}.
\tag{11}
$$

Meanwhile, setting $t = 1$ in Equation (10) yields

$$
\mathbf{\Delta}_{\text{total},1}^* = (\mathbf{V}_1 - \mathbf{W}_0\mathbf{K}_1)\mathbf{K}_1^\top\mathbf{P}\left(\mathbf{K}_1\mathbf{K}_1^\top\mathbf{P} + \mathbf{I}\right)^{-1}.
\tag{12}
$$

Comparing Equations (11) and (12), we have $\boldsymbol{\Delta}_1^* = \boldsymbol{\Delta}_{\text{total},1}^*$.

Next, we assume that the statement is true for $t = T - 1$, and show that it also holds for $t = T$, for any $T > 2$. Taking $t = T$ in Equation (9), we have the following relation:

$$\boldsymbol{\Delta}_T^* \left( \mathbf{K}_{1:T-1} \mathbf{K}_{1:T-1}^\top \mathbf{P} + \mathbf{K}_T \mathbf{K}_T^\top \mathbf{P} + \mathbf{I} \right) = \left( \mathbf{V}_T - \mathbf{W}_{T-1} \mathbf{K}_T \right) \mathbf{K}_T^\top \mathbf{P}. \tag{13}$$

We then rewrite the left-hand side of Equation (13) using Equation (8):

$$\boldsymbol{\Delta}_T^* \left( \mathbf{K}_{1:T-1} \mathbf{K}_{1:T-1}^\top \mathbf{P} + \mathbf{K}_T \mathbf{K}_T^\top \mathbf{P} + \mathbf{I} \right) = \boldsymbol{\Delta}_T^* \left( \sum_{\tau=1}^{T} \mathbf{K}_\tau \mathbf{K}_\tau^\top \mathbf{P} + \mathbf{I} \right),$$

and rewrite the right-hand side of Equation (13) using Equation (6):

$$\left( \mathbf{V}_T - \mathbf{W}_{T-1} \mathbf{K}_T \right) \mathbf{K}_T^\top \mathbf{P} = \left( \mathbf{V}_T - \left( \mathbf{W}_0 + \sum_{\tau=1}^{T-1} \boldsymbol{\Delta}_\tau^* \right) \mathbf{K}_T \right) \mathbf{K}_T^\top \mathbf{P}$$

$$= \mathbf{R}_T' \mathbf{K}_T^\top \mathbf{P} - \left( \sum_{\tau=1}^{T-1} \boldsymbol{\Delta}_\tau^* \right) \mathbf{K}_T \mathbf{K}_T^\top \mathbf{P}$$

Thus, we arrive at

$$\boldsymbol{\Delta}_T^* \left( \sum_{\tau=1}^{T} \mathbf{K}_\tau \mathbf{K}_\tau^\top \mathbf{P} + \mathbf{I} \right) = \mathbf{R}_T' \mathbf{K}_T^\top \mathbf{P} - \left( \sum_{\tau=1}^{T-1} \boldsymbol{\Delta}_\tau^* \right) \mathbf{K}_T \mathbf{K}_T^\top \mathbf{P}. \tag{14}$$

Further, by our induction assumption for $t = T - 1$, we have the following equality:

$$\left( \sum_{\tau=1}^{T-1} \boldsymbol{\Delta}_\tau^* \right) \left( \sum_{\tau=1}^{T-1} \mathbf{K}_\tau \mathbf{K}_\tau^\top \mathbf{P} + \mathbf{I} \right) = \sum_{\tau=1}^{T-1} \mathbf{R}_\tau' \mathbf{K}_\tau^\top \mathbf{P} \tag{15}$$

Adding both sides of Equation (14) and Equation (15) and rearranging the terms, we have

$$\left( \sum_{\tau=1}^{T} \boldsymbol{\Delta}_\tau^* \right) \left( \sum_{\tau=1}^{T} \mathbf{K}_\tau \mathbf{K}_\tau^\top \mathbf{P} + \mathbf{I} \right) = \sum_{\tau=1}^{T} \mathbf{R}_\tau' \mathbf{K}_\tau^\top \mathbf{P},$$

indicating that $\sum_{\tau=1}^{T} \boldsymbol{\Delta}_\tau^* = \boldsymbol{\Delta}_{\text{total},T}^*$. This completes the proof. $\qquad\square$

### A.3. Generalized Equivalence of One-Time and Sequential Editing

By reintroducing the neglected term and adding an adjustable coefficient for $L^2$ regularization, we restore AlphaEdit's full optimization target as follows:

$$\arg\min_{\boldsymbol{\Delta}_t} \|(\mathbf{W}_{t-1} + \boldsymbol{\Delta}_t \mathbf{P}) \mathbf{K}_t - \mathbf{V}_t\|_F^2 + \|\boldsymbol{\Delta}_t \mathbf{P} \mathbf{K}_{1:t-1}\|_F^2 + \|\boldsymbol{\Delta}_t \mathbf{P} \mathbf{K}_0\|_F^2 + \lambda \|\boldsymbol{\Delta}_t \mathbf{P}\|_F^2,$$

where the term $\|\boldsymbol{\Delta}_t \mathbf{P} \mathbf{K}_0\|_F^2$ was originally neglected due to null-space projection. This full optimization target results in the following updating rule for $\boldsymbol{\Delta}_t^*$:

$$\boldsymbol{\Delta}_t^* = \left( \mathbf{V}_t - \mathbf{W}_{t-1} \mathbf{K}_t \right) \mathbf{K}_t^\top \mathbf{P} \left( \mathbf{K}_{1:t-1} \mathbf{K}_{1:t-1}^\top \mathbf{P} + \mathbf{K}_t \mathbf{K}_t^\top \mathbf{P} + \mathbf{K}_0 \mathbf{K}_0^\top \mathbf{P} + \lambda \mathbf{I} \right)^{-1},$$

This leads to Proposition 3.2 restated as follows:

**Proposition** (Generalized Equivalence of One-Time and Sequential Editing). *Consider a one-time editing objective that jointly incorporates all knowledge updates collected from edit steps $1$ through $t$. The resulting batch solution is given by*

$$\boldsymbol{\Delta}_{total,t}^* = \sum_{\tau=1}^{t} \mathbf{R}_\tau' \mathbf{K}_\tau^\top \mathbf{P} \left( \mathbf{K}_0 \mathbf{K}_0^\top \mathbf{P} + \sum_{\tau=1}^{t} \mathbf{K}_\tau \mathbf{K}_\tau^\top \mathbf{P} + \lambda \mathbf{I} \right)^{-1}.$$

*Let the sequential update at step $t$ be defined as*

$$\boldsymbol{\Delta}_t^* = \mathbf{R}_t' \mathbf{K}_t^\top \mathbf{P} \left( \mathbf{K}_0 \mathbf{K}_0^\top \mathbf{P} + \sum_{\tau=1}^t \mathbf{K}_\tau \mathbf{K}_\tau^\top \mathbf{P} + \lambda \mathbf{I} \right)^{-1}.$$

*Then, for any choice of $\mathbf{P}$ and $\lambda \geq 0$, the accumulated sequential updates satisfy*

$$\sum_{\tau=1}^t \boldsymbol{\Delta}_\tau^* = \boldsymbol{\Delta}_{total,t}^*.$$

*Proof.* Using induction and following the proof of Lemma 3.1, we note that the argument does not rely on any specific property of the null-space projection. Therefore, this equivalence holds generally for any choice of $\mathbf{P}$. $\square$

Since $\boldsymbol{\Delta}_{\text{total},t}^*$ solves a valid OTE target for any choice of $\mathbf{P}$, and in particular when $\mathbf{P} = \mathbf{I}$, we obtain the sequential MEMIT algorithm. This leads to the following remark:

*Remark* A.1. Sequential MEMIT admits the following update rule:

$$\boldsymbol{\Delta}_t^* = \mathbf{R}_t \mathbf{K}_t^\top \left( \mathbf{K}_0 \mathbf{K}_0^\top + \sum_{\tau=1}^t \mathbf{K}_\tau \mathbf{K}_\tau^\top \right)^{-1}.$$

## A.4. Constructed Sequential Updates as OLS Solutions

**Proposition** (Constructed Sequential Updates as OLS Solutions). *Consider a sequential editing task with steps $t = 1, 2, \ldots, T$. Let the one-time editing (OTE) objective at step $t$ be*

$$\boldsymbol{\Delta}_{\text{total},t}^* = \arg\min_{\boldsymbol{\Delta}} \ \mathcal{L}_t(\boldsymbol{\Delta}) + \mathcal{R}(\boldsymbol{\Delta}),$$

*where $\mathcal{R}$ is a closed convex regularization function.*

*Define the sequential update at step $t$ via the difference of consecutive OTE solutions:*

$$\tilde{\boldsymbol{\Delta}}_t := \boldsymbol{\Delta}_{\text{total},t}^* - \boldsymbol{\Delta}_{\text{total},t-1}^*, \ \ \text{with } \boldsymbol{\Delta}_{\text{total},0}^* := \mathbf{0}.$$

*Assume that the loss $\mathcal{L}_t(\cdot)$ satisfies the following **shifted quadratic representability** condition: for any $t$ and $\boldsymbol{\Delta}$,*

$$\mathcal{L}_t(\boldsymbol{\Delta}_{\text{total},t-1}^* + \boldsymbol{\Delta}) = \ell_t(\boldsymbol{\Delta}) + \langle \nabla \mathcal{L}_t(\boldsymbol{\Delta}_{\text{total},t-1}^*), \boldsymbol{\Delta} \rangle + c_t,$$

*where:*

- *$\ell_t(\boldsymbol{\Delta})$ is a convex quadratic function of $\boldsymbol{\Delta}$,*

- *$c_t \in \mathbb{R}$ is a constant independent of $\boldsymbol{\Delta}$,*

- *$\nabla \mathcal{L}_t$ denotes the Fréchet derivative of $\mathcal{L}_t$.*

*Then, the sequential update $\tilde{\boldsymbol{\Delta}}_t$ is the unique solution to the following **sequential optimization problem**:*

$$\boldsymbol{\Delta}_t^* = \arg\min_{\boldsymbol{\Delta}} \ \ell_t(\boldsymbol{\Delta}) + \langle \nabla \mathcal{L}_t(\boldsymbol{\Delta}_{\text{total},t-1}^*), \boldsymbol{\Delta} \rangle + \mathcal{R}(\boldsymbol{\Delta}_{\text{total},t-1}^* + \boldsymbol{\Delta}).$$

*Proof.* Let $\boldsymbol{\Delta}_{\text{total},t}^*$ be the solution of the OTE problem:

$$\boldsymbol{\Delta}_{\text{total},t}^* = \arg\min_{\boldsymbol{\Delta}} \ \mathcal{L}_t(\boldsymbol{\Delta}) + \mathcal{R}(\boldsymbol{\Delta}). \tag{16}$$

By the first-order optimality condition for convex composite functions, we have:

$$\nabla \mathcal{L}_t(\boldsymbol{\Delta}_{\text{total},t}^*) + \partial \mathcal{R}(\boldsymbol{\Delta}_{\text{total},t}^*) \ni \mathbf{0}, \tag{17}$$

where $\partial\mathcal{R}$ denotes the subdifferential of $\mathcal{R}$.

Now, write $\boldsymbol{\Delta}^*_{\text{total},t} = \boldsymbol{\Delta}^*_{\text{total},t-1} + \boldsymbol{\Delta}_t$. Using the shifted quadratic representability assumption, we expand $\mathcal{L}_t$:

$$\mathcal{L}_t(\boldsymbol{\Delta}^*_{\text{total},t}) = \ell_t(\boldsymbol{\Delta}_t) + \langle\nabla\mathcal{L}_t(\boldsymbol{\Delta}^*_{\text{total},t-1}), \boldsymbol{\Delta}_t\rangle + c_t. \tag{18}$$

The gradient of $\mathcal{L}_t$ at $\boldsymbol{\Delta}^*_{\text{total},t}$ can be computed from (18):

$$\nabla\mathcal{L}_t(\boldsymbol{\Delta}^*_{\text{total},t}) = \nabla\ell_t(\boldsymbol{\Delta}_t) + \nabla\mathcal{L}_t(\boldsymbol{\Delta}^*_{\text{total},t-1}). \tag{19}$$

Substituting (19) into the optimality condition (17) yields:

$$\nabla\ell_t(\boldsymbol{\Delta}_t) + \nabla\mathcal{L}_t(\boldsymbol{\Delta}^*_{\text{total},t-1}) + \partial\mathcal{R}(\boldsymbol{\Delta}^*_{\text{total},t-1} + \boldsymbol{\Delta}_t) \ni \mathbf{0}. \tag{20}$$

Observe that (20) is exactly the first-order optimality condition for the sequential problem:

$$\min_{\boldsymbol{\Delta}} \; \ell_t(\boldsymbol{\Delta}) + \langle\nabla\mathcal{L}_t(\boldsymbol{\Delta}^*_{\text{total},t-1}), \boldsymbol{\Delta}\rangle + \mathcal{R}(\boldsymbol{\Delta}^*_{\text{total},t-1} + \boldsymbol{\Delta}). \tag{21}$$

Since $\ell_t$ is strictly convex (as a positive definite quadratic form) and $\mathcal{R}$ is convex, the solution to (21) is unique. Thus, $\boldsymbol{\Delta}_t$ satisfies (20) and therefore is the unique minimizer of (21).

This completes the proof. $\square$

**Corollary A.2** (Ordinary Least Squares Case). *If $\mathcal{L}_t(\boldsymbol{\Delta}) = \|(\mathbf{W} + \boldsymbol{\Delta})\mathbf{K}_{0:t} - \mathbf{V}_{0:t}\|^2_F$ and $\mathcal{R}(\boldsymbol{\Delta}) = \lambda\|\boldsymbol{\Delta}\|^2_F$, then the shifted quadratic representability condition holds with*

$$\ell_t(\boldsymbol{\Delta}) = \|\boldsymbol{\Delta}\mathbf{K}_{0:t}\|^2_F, \quad \textit{and} \quad \nabla\mathcal{L}_t(\boldsymbol{\Delta}^*_{\text{total},t-1}) = 2\big((\mathbf{W} + \boldsymbol{\Delta}^*_{\text{total},t-1})\mathbf{K}_{0:t} - \mathbf{V}_{0:t}\big)\mathbf{K}^\top_{0:t}. \tag{22}$$

*Consequently, the sequential update $\boldsymbol{\Delta}_t$ solves the problem:*

$$\min_{\boldsymbol{\Delta}} \|(\mathbf{W}_{t-1} + \boldsymbol{\Delta})\mathbf{K}_{0:t} - \mathbf{V}_{0:t}\|^2_F + \lambda\|\boldsymbol{\Delta}^*_{\text{total},t-1} + \boldsymbol{\Delta}\|^2_F, \tag{23}$$

*where $\mathbf{W}_{t-1} = \mathbf{W} + \boldsymbol{\Delta}^*_{\text{total},t-1}$.*

*Proof.* Direct computation yields:

$$\begin{aligned}
\mathcal{L}_t(\boldsymbol{\Delta}^*_{\text{total},t-1} + \boldsymbol{\Delta}) &= \|(\mathbf{W}_{t-1} + \boldsymbol{\Delta})\mathbf{K}_{0:t} - \mathbf{V}_{0:t}\|^2_F \\
&= \|\boldsymbol{\Delta}\mathbf{K}_{0:t}\|^2_F + 2\langle(\mathbf{W}_{t-1}\mathbf{K}_{0:t} - \mathbf{V}_{0:t})\mathbf{K}^\top_{0:t}, \boldsymbol{\Delta}\rangle + \text{const}.
\end{aligned} \tag{24}$$

The expansion in (24) matches the shifted quadratic representability condition, hence the sequential update $\boldsymbol{\Delta}_t$ indeed solves (23). $\square$

## A.5. Error Correction for Post-processing Regularization

In this section, we derive the error-correction term for post-processing regularization. Formally, we have the following proposition:

**Proposition A.3.** *At any step $t \geq 1$, iteratively applying Algorithm 1 to obtain $\boldsymbol{\Delta}_t$ and $\mathbf{W}^{\mathcal{R}}_{t-1}$, we have $\mathbf{W}^{\mathcal{R}}_{t-1} + \boldsymbol{\Delta}_t = \mathbf{W}_0 + \boldsymbol{\Delta}_{\text{total},t}$.*

*Proof.* Consider the following alternative optimization target. We first show that Algorithm 1 effectively solves the following target by induction:

$$\arg\min_{\boldsymbol{\Delta}_t} \sum_{\tau=0}^{t} \left\|\mathbf{V}_\tau - \left(\mathbf{W}^{\mathcal{R}}_{t-1} + \boldsymbol{\Delta}_t\right)\mathbf{K}_\tau\right\|^2_F, \tag{25}$$

where $\mathbf{W}_t^{\mathcal{R}} = \mathbf{W}_0 + \sum_{\tau=1}^{t} \mathcal{R}_p(\boldsymbol{\Delta}_\tau)$. The base step is trivial to show: we just substitute $t = 1$ and we can verify that $\boldsymbol{\Delta}_1$ aligns with what we obtained from Algorithm 1 at $t = 1$. The inductive assumption at $t = T - 1$ then gives us the following, which is the normal equation of the target stated in Equation (25):

$$\boldsymbol{\Delta}_{T-1}^* \sum_{\tau=0}^{T-1} \mathbf{K}_\tau \mathbf{K}_\tau^\top = \sum_{\tau=0}^{T-1} \left( \mathbf{V}_\tau - \mathbf{W}_{T-2}^{\mathcal{R}} \mathbf{K}_\tau \right) \mathbf{K}_\tau^\top \tag{26}$$

We then show that Algorithm 1 solves the following target at $t = T$:

$$\arg \min_{\boldsymbol{\Delta}_T} \sum_{\tau=0}^{T} \left\| \mathbf{V}_\tau - \left( \mathbf{W}_{T-1}^{\mathcal{R}} + \boldsymbol{\Delta}_T \right) \mathbf{K}_\tau \right\|_F^2 . \tag{27}$$

This requires $\boldsymbol{\Delta}_T^*$ obtained from Algorithm 1 to satisfy the following normal equation:

$$\boldsymbol{\Delta}_T^* \sum_{\tau=0}^{T} \mathbf{K}_\tau \mathbf{K}_\tau^\top = \sum_{\tau=0}^{T} \left( \mathbf{V}_\tau - \mathbf{W}_{T-1}^{\mathcal{R}} \mathbf{K}_\tau \right) \mathbf{K}_\tau^\top .$$

Comparing to the iterative solution in Algorithm 1, which can be equivalently expressed as follows:

$$\boldsymbol{\Delta}_T^* \sum_{\tau=0}^{T} \mathbf{K}_\tau \mathbf{K}_\tau^\top = \left( \mathbf{V}_T - \mathbf{W}_{T-1}^{\mathcal{R}} \mathbf{K}_T \right) \mathbf{K}_T^\top - \mathbf{E}_{T-1},$$

we can see that it suffices to show that $\mathbf{E}_{T-1} = -\sum_{\tau=0}^{T-1} \left( \mathbf{V}_\tau - \mathbf{W}_{T-1}^{\mathcal{R}} \mathbf{K}_\tau \right) \mathbf{K}_\tau^\top$. We approach this by expanding the right-hand side as follows:

$$\sum_{\tau=0}^{T-1} \left( \mathbf{V}_\tau - \mathbf{W}_{T-1}^{\mathcal{R}} \mathbf{K}_\tau \right) \mathbf{K}_\tau^\top = \sum_{\tau=0}^{T-1} \left( \mathbf{V}_\tau - \mathbf{W}_{T-2}^{\mathcal{R}} \mathbf{K}_\tau \right) \mathbf{K}_\tau^\top - \mathcal{R}_p(\boldsymbol{\Delta}_{T-1}^*) \sum_{\tau=0}^{T-1} \mathbf{K}_\tau \mathbf{K}_\tau^\top \tag{28}$$

Substituting Equation (26) into Equation (28) allows us to complete the induction.

Additionally, at step $t$, the optimization target of a canonical OTE problem admits the following form:

$$\arg \min_{\boldsymbol{\Delta}_{\text{total},t}} \sum_{\tau=0}^{t} \left\| \mathbf{V}_\tau - \left( \mathbf{W}_0 + \boldsymbol{\Delta}_{\text{total},t} \right) \mathbf{K}_\tau \right\|_F^2 .$$

Due to the uniqueness of the optimum of OLS, we complete the proof. $\qquad\square$

### A.6. Conflict Resolving and Selective Forgetting

In this section, we provide a more detailed discussion of conflict resolving in SE, adding more information to Section 3.4.

At editing step $t$, denote the new knowledge set as $\mathcal{A}_t = \{(\mathbf{k}, \mathbf{v}) | \mathbf{k} \in \mathcal{K}_t, \mathbf{v} \in \mathcal{V}_t\}$. Denote all the previously edited and preserved knowledge set as $\mathcal{P}_{t-1}$. Then, some keys in $\mathcal{A}_t$ and $\mathcal{P}_{t-1}$ may overlap, regardless of their paired values, and we must resolve these values to ensure reliable editing. Denote this overlap set as $\mathcal{K}_o = \{\mathbf{k} | (\mathbf{k}, \mathbf{v}) \in \mathcal{A}_t\} \cap \{\mathbf{k} | (\mathbf{k}, \mathbf{v}') \in \mathcal{P}_{t-1}\}$. If $\mathbf{v} = \mathbf{v}'$, we can simply overwrite $\mathbf{v}'$ with $\mathbf{v}$. If $\mathbf{v} \neq \mathbf{v}'$, we need to remove the old value $\mathbf{v}'$ and insert the new value $\mathbf{v}$. We denote the function for this resolving process as $\text{Resolve}(\cdot)$. We can then form three disjoint sets as follows:

1. The resolved set from the overlapping region: $\mathcal{B}_o^{(t)} = \{(\mathbf{k}, \text{Resolve}(\mathbf{v}, \mathbf{v}')) | \mathbf{k} \in \mathcal{K}_o\}$;

2. The non-overlapping set from $\mathcal{A}_t$: $\mathcal{B}_{\mathcal{A}_t/\mathcal{P}_{t-1}}^{(t)} = \{(\mathbf{k}, \mathbf{v}) \in \mathcal{A}_t | \mathbf{k} \notin \mathcal{K}_{\mathcal{P}_{t-1}}\}$;

3. The non-overlapping set from $\mathcal{P}_{t-1}$: $\mathcal{B}_{\mathcal{P}_{t-1}/\mathcal{A}_t}^{(t)} = \{(\mathbf{k}, \mathbf{v}) \in \mathcal{P}_{t-1} | \mathbf{k} \notin \mathcal{K}_{\mathcal{A}_t}\}$;

where $\mathcal{K}_{\mathcal{P}_{t-1}}$ is the set of keys in $\mathcal{P}_{t-1}$, and $\mathcal{K}_{\mathcal{A}_t}$ is the set of keys in $\mathcal{A}_t$. The total fact set after resolving the conflict is then formed by

$$\mathcal{P}_t = \mathcal{B}_{\mathrm{o}}^{(t)} \cup \mathcal{B}_{\mathcal{A}_t/\mathcal{P}_{t-1}}^{(t)} \cup \mathcal{B}_{\mathcal{P}_{t-1}/\mathcal{A}_t}^{(t)}. \tag{29}$$

We then find the perturbation to the weight matrix to associate the keys and values from the resolved knowledge set $\mathcal{P}_t$. We have the following proposition (which is an expanded version of the original Proposition 3.5 in the main text):

**Proposition.** *Given the previous edit perturbations $\boldsymbol{\Delta}_1^*, \boldsymbol{\Delta}_2^*, \ldots, \boldsymbol{\Delta}_{t-1}^*$, the $t$-step perturbation for SPF is*

$$\boldsymbol{\Delta}_t^* = \left\{ \left[ \mathbf{V}_t - \left( \mathbf{W} + \sum_{\tau=1}^{t-1} \boldsymbol{\Delta}_\tau^* \right) \mathbf{K}_t \right] \mathbf{K}_t^\top \right.$$
$$\left. - \left[ \mathbf{V}_{\mathcal{B}_o^{(t)}} - \left( \mathbf{W} + \sum_{\tau=1}^{t-1} \boldsymbol{\Delta}_\tau^* \right) \mathbf{K}_{\mathcal{B}_o^{(t)}} \right) \right] \mathbf{K}_{\mathcal{B}_o^{(t)}}^\top \right\}$$
$$\cdot \left( \mathbf{K}_{\mathcal{P}_{t-1}} \mathbf{K}_{\mathcal{P}_{t-1}}^\top + \mathbf{K}_t \mathbf{K}_t^\top - \mathbf{K}_{\mathcal{B}_o^{(t)}} \mathbf{K}_{\mathcal{B}_o^{(t)}}^\top \right)^{-1}, \tag{30}$$

*where $\mathbf{K}_{\mathcal{P}_{t-1}}, \mathbf{V}_{\mathcal{P}_{t-1}}$ are stacked keys and values from $\mathcal{P}_{t-1}$; $\mathbf{K}_t, \mathbf{V}_t$ are stacked keys and values from $\mathcal{A}_t$, and $\mathbf{K}_{\mathcal{B}_o^{(t)}}, \mathbf{V}_{\mathcal{B}_o^{(t)}}$ are stacked keys and values from $\mathcal{B}_o^{(t)}$, respectively.*

*Proof.* For notation simplicity, denote $\mathbf{K}_{A/P}, \mathbf{V}_{A/P}$ as stacked keys and values from $\mathcal{B}_{\mathcal{A}_t/\mathcal{P}_{t-1}}^{(t)}$; $\mathbf{K}_{P/A}, \mathbf{V}_{P/A}$ as stacked keys and values from $\mathcal{B}_{\mathcal{P}_{t-1}/\mathcal{A}_t}^{(t)}$, and $\mathbf{K}_{AP}, \mathbf{V}_{AP}$ as stacked keys and values from $\mathcal{B}_o^{(t)}$, respectively. Then the objective function for the $t$-th edit is

$$\boldsymbol{\Delta}_t^* = \underset{\tilde{\boldsymbol{\Delta}}_t}{\arg\min} \left\{ \left\| \left( \mathbf{W} + \sum_{\tau=1}^{t-1} \boldsymbol{\Delta}_\tau^* + \tilde{\boldsymbol{\Delta}}_t \right) \mathbf{K}_{A/P} - \mathbf{V}_{A/P} \right\|_F^2 \right.$$
$$\left\| \left( \mathbf{W} + \sum_{\tau=1}^{t-1} \boldsymbol{\Delta}_\tau^* + \tilde{\boldsymbol{\Delta}}_t \right) \mathbf{K}_{P/A} - \mathbf{V}_{P/A} \right\|_F^2$$
$$\left. \left\| \left( \mathbf{W} + \sum_{\tau=1}^{t-1} \boldsymbol{\Delta}_\tau^* + \tilde{\boldsymbol{\Delta}}_t \right) \mathbf{K}_{AP} - \mathbf{V}_{AP} \right\|_F^2 \right\}.$$

With the first-order gradient condition, it is easy to get

$$\boldsymbol{\Delta}_t^* = \left\{ \left[ \mathbf{V}_{A/P} - \left( \mathbf{W} + \sum_{\tau=1}^{t-1} \boldsymbol{\Delta}_\tau^* \right) \mathbf{K}_{A/P} \right] \mathbf{K}_{A/P}^\top + \left[ \mathbf{V}_{P/A} - \left( \mathbf{W} + \sum_{\tau=1}^{t-1} \boldsymbol{\Delta}_\tau^* \right) \mathbf{K}_{P/A} \right] \mathbf{K}_{P/A}^\top \right.$$
$$\left. + \left[ \mathbf{V}_{AP} - \left( \mathbf{W} + \sum_{\tau=1}^{t-1} \boldsymbol{\Delta}_\tau^* \right) \mathbf{K}_{AP} \right] \mathbf{K}_{AP}^\top \right\} \cdot \left( \mathbf{K}_{A/P} \mathbf{K}_{A/P}^\top + \mathbf{K}_{P/A} \mathbf{K}_{P/A}^\top + \mathbf{K}_{AP} \mathbf{K}_{AP}^\top \right)^{-1}.$$

Rearrange $\mathbf{K}_t, \mathbf{V}_t$ from $\mathcal{A}_t$ as $\mathbf{K}_t = [\mathbf{K}_{A/P} \quad \mathbf{K}_{AP}], \mathbf{V}_t = [\mathbf{V}_{A/P} \quad \mathbf{V}_{AP}]$, and rearrange $\mathbf{K}_{\mathcal{P}_{t-1}}, \mathbf{V}_{\mathcal{P}_{t-1}}$ from $\mathcal{P}_{t-1}$ as $\mathbf{K}_{\mathcal{P}_{t-1}} = [\mathbf{K}_{P/A} \quad \mathbf{K}_{AP}], \mathbf{V}_{\mathcal{P}_{t-1}} = [\mathbf{V}_{P/A} \quad \mathbf{V}_{AP}]$. Further, denote $\mathbf{W}_{t-1} = \mathbf{W} + \sum_{\tau=1}^{t-1} \boldsymbol{\Delta}_\tau^*$. We can see that

$$\left( \mathbf{V}_t - \mathbf{W}_{t-1} \mathbf{K}_t \right) \mathbf{K}_t^\top + \left( \mathbf{V}_{\mathcal{P}_{t-1}} - \mathbf{W}_{t-1} \mathbf{K}_{\mathcal{P}_{t-1}} \right) \mathbf{K}_{\mathcal{P}_{t-1}}^\top$$
$$= \left\{ [\mathbf{V}_{A/P} \quad \mathbf{V}_{AP}] - \mathbf{W}_{t-1}[\mathbf{K}_{A/P} \quad \mathbf{K}_{AP}] \right\} \begin{bmatrix} \mathbf{K}_{A/P}^\top \\ \mathbf{K}_{AP}^\top \end{bmatrix} + \left\{ [\mathbf{V}_{P/A} \quad \mathbf{V}_{AP}] - \mathbf{W}_{t-1}[\mathbf{K}_{P/A} \quad \mathbf{K}_{AP}] \right\} \begin{bmatrix} \mathbf{K}_{P/A}^\top \\ \mathbf{K}_{AP}^\top \end{bmatrix}$$
$$= \left[ \mathbf{V}_{A/P} - \mathbf{W}_{t-1} \mathbf{K}_{A/P} \right] \mathbf{K}_{A/P}^\top + \left[ \mathbf{V}_{P/A} - \mathbf{W}_{t-1} \mathbf{K}_{P/A} \right] \mathbf{K}_{P/A}^\top + 2 \left[ \mathbf{V}_{AP} - \mathbf{W}_{t-1} \mathbf{K}_{AP} \right] \mathbf{K}_{AP}^\top.$$

Similarly, we can get

$$\mathbf{K}_t \mathbf{K}_t^\top + \mathbf{K}_{\mathcal{P}_{t-1}} \mathbf{K}_{\mathcal{P}_{t-1}}^\top = \mathbf{K}_{A/P} \mathbf{K}_{A/P}^\top + \mathbf{K}_{P/A} \mathbf{K}_{P/A}^\top + 2 \mathbf{K}_{AP} \mathbf{K}_{AP}^\top.$$

Thus, we have

$$\boldsymbol{\Delta}_t^* = \left[ \left( \mathbf{V}_t - \mathbf{W}_{t-1} \mathbf{K}_t \right) \mathbf{K}_t^\top + \left( \mathbf{V}_{\mathcal{P}_{t-1}} - \mathbf{W}_{t-1} \mathbf{K}_{\mathcal{P}_{t-1}} \right) \mathbf{K}_{\mathcal{P}_{t-1}}^\top - \left( \mathbf{V}_{AP} - \mathbf{W}_{t-1} \mathbf{K}_{AP} \right) \mathbf{K}_{AP}^\top \right]$$
$$\cdot \left( \mathbf{K}_t \mathbf{K}_t^\top + \mathbf{K}_{\mathcal{P}_{t-1}} \mathbf{K}_{\mathcal{P}_{t-1}}^\top - \mathbf{K}_{AP} \mathbf{K}_{AP}^\top \right)^{-1}.$$

Since $\mathbf{W}_{t-1} = \mathbf{W} + \sum_{\tau=1}^{t-1} \boldsymbol{\Delta}_\tau^*$ is the weight matrix from the previous edit step, we have $\left( \mathbf{V}_{\mathcal{P}_{t-1}} - \mathbf{W}_{t-1} \mathbf{K}_{\mathcal{P}_{t-1}} \right) \mathbf{K}_{\mathcal{P}_{t-1}}^\top = 0$. Also by definition, $\mathbf{K}_{AP} = \mathbf{K}_{\mathcal{B}_o^{(t)}}$ and $\mathbf{V}_{AP} = \mathbf{V}_{\mathcal{B}_o^{(t)}}$, so we get Equation (30). $\square$

Comparing Equation (30) with conclusion of Remark A.1, we can see that at each step $t$, instead of simply inserting the new knowledge $\mathcal{A}_t$ into the previous knowledge $\mathcal{P}_{t-1}$, we resolve the contradictory facts by subtracting the terms related to the overlapping set (i.e., $\mathcal{B}_{\mathrm{o}}^{(t)}$) from the two parts of the formula. This is intuitive in the sense that if we preserve all the knowledge without resolving the overlapping set, then the contradictory facts will be counted into the OLS solution, leading to unreliable and unstable edits.

To further mention an important special case where the LLM only memorizes the latest knowledge, we have the following remark for the memorize-the-latest task mentioned in Section 3.1, where we iteratively memorize the latest knowledge while forgetting knowledge in the previous edit.

*Remark* A.4. The solution to the *Memorize-the-Latest* task we introduced in Section 3.1 is a special case for Equation (30) where

$$\mathbf{K}_{\mathcal{P}_{t-1}} = [\mathbf{K}_0 \mid \mathbf{K}_{t-1}] \quad \text{and} \quad \mathbf{K}_{\mathcal{B}_{\mathrm{o}}^{(t)}} = \mathbf{K}_{t-1}.$$

This allows us to write the following updating rule from Equation (30) for the Memorize-the-Latest task as follows:

$$\boldsymbol{\Delta}_t^* \left( \mathbf{K}_0 \mathbf{K}_0^\top + \mathbf{K}_t \mathbf{K}_t^\top \right) = \left( \mathbf{V}_t - \mathbf{W}_{t-1} \mathbf{K}_t \right) \mathbf{K}_t^\top - \left( \mathbf{V}_{t-1} - \mathbf{W}_{t-1} \mathbf{K}_{t-1} \right) \mathbf{K}_{t-1}^\top \tag{31}$$

At first glance, this update seems different from what we obtain by directly solving the following optimization target:

$$\boldsymbol{\Delta}_t^* = \arg \min_{\boldsymbol{\Delta}_t} \| (\mathbf{W}_{t-1} + \boldsymbol{\Delta}_t)[\mathbf{K}_0 \mid \mathbf{K}_t] - [\mathbf{V}_0 \mid \mathbf{V}_t] \|_F^2 ,$$

whose solution admits the following equality:

$$\boldsymbol{\Delta}_t^* \left( \mathbf{K}_0 \mathbf{K}_0^\top + \mathbf{K}_t \mathbf{K}_t^\top \right) = \left( \mathbf{V}_t - \mathbf{W}_{t-1} \mathbf{K}_t \right) \mathbf{K}_t^\top + \left( \mathbf{V}_0 - \mathbf{W}_{t-1} \mathbf{K}_0 \right) \mathbf{K}_0^\top . \tag{32}$$

However, since at step $t-1$, $\mathbf{W}_{t-1}$ solves the following objective in the Memorize-the-Latest task:

$$\arg \min_{\tilde{\mathbf{W}}_{t-1}} \| \tilde{\mathbf{W}}_{t-1}[\mathbf{K}_0|\mathbf{K}_{t-1}] - [\mathbf{V}_0|\mathbf{V}_{t-1}] \|_F^2 .$$

The corresponding normal equation gives us

$$(\mathbf{V}_0 - \mathbf{W}_{t-1}\mathbf{K}_0)\mathbf{K}_0^\top + (\mathbf{V}_{t-1} - \mathbf{W}_{t-1}\mathbf{K}_{t-1})\mathbf{K}_{t-1}^\top = 0.$$

This shows that Equations (31) and (32) are equivalent.

## B. Experimental Setup

### B.1. Dataset

We evaluate editing performance on two widely used benchmarks, CounterFact and ZsRE, which jointly probe edit *efficacy*, *generalization*, and *specificity/locality*.

**CounterFact (Meng et al., 2022).** CounterFact is designed to stress-test factual editing under counterfactual settings, and is typically considered more challenging than relation-style QA benchmarks. It pairs factual and counterfactual assertions, and constructs *out-of-scope* (OOS) queries by swapping the subject entity with a semantically similar (approximate) entity while keeping the predicate/relation unchanged, thereby encouraging the editor to be precise rather than over-generalize. Following common model-editing protocols, it supports measurements analogous to those used on ZsRE (edit success, generalization to paraphrases, and locality/specificity on unrelated prompts). In addition, CounterFact provides multiple meaning-preserving generation prompts per case, enabling evaluation of the generated continuations with respect to fluency and semantic consistency across prompts.

**ZsRE (Levy et al., 2017).** ZsRE is a question-answering benchmark originally introduced for zero-shot relation extraction framed as reading comprehension. In the editing setting, each instance specifies a subject string and target answer(s) as the desired edit. The dataset includes paraphrased questions—commonly produced via back-translation—as *equivalent neighbors* for assessing generalization, and uses natural questions as OOS queries to quantify locality/specificity.

## B.2. Metrics

We summarize the evaluation metrics used for CounterFact and ZsRE.

### B.2.1. COUNTERFACT METRICS

Following prior work (Meng et al., 2022; 2023), we report:

**Efficacy (efficacy success).** The fraction of edit cases for which the edited target becomes more probable than the original completion under the original prompt, with $(s, r, o^c)$ representing the original correct knowledge and $(s, r, o)$ representing the edited counterfactual knowledge:

$$\text{Eff}_{\text{CF}} = \mathbb{E}_i \Big[ P_{f_\theta}(o_i \mid (s_i, r_i)) > P_{f_\theta}(o_i^c \mid (s_i, r_i)) \Big]. \tag{33}$$

**Generalization (paraphrase success).** The fraction of cases where the edited target is preferred over the original completion on meaning-equivalent paraphrases $N((s_i, r_i))$:

$$\text{Gen}_{\text{CF}} = \mathbb{E}_i \Big[ P_{f_\theta}(o_i \mid N((s_i, r_i))) > P_{f_\theta}(o_i^c \mid N((s_i, r_i))) \Big]. \tag{34}$$

**Specificity (neighborhood success).** CounterFact additionally evaluates locality using *neighborhood* prompts $O((s_i, r_i))$ that are semantically related but refer to distinct subjects. We measure the fraction of neighborhood prompts for which the model assigns higher probability to the correct neighborhood fact:

$$\text{Spec}_{\text{CF}} = \mathbb{E}_i \Big[ P_{f_\theta}(o_i \mid O((s_i, r_i))) > P_{f_\theta}(o_i^c \mid O((s_i, r_i))) \Big]. \tag{35}$$

**Fluency (generation entropy).** To detect degenerate generations with excessive repetition, we compute an $n$-gram entropy score over model outputs. Let $g_n(\cdot)$ be the empirical frequency distribution of $n$-grams; the fluency score is

$$\text{Fluency} = -\frac{2}{3} \sum_k g_2(k) \log_2 g_2(k) + \frac{4}{3} \sum_k g_3(k) \log_2 g_3(k). \tag{36}$$

**Consistency (reference similarity).** We assess whether generated text remains semantically aligned with an external reference by computing the cosine similarity between TF–IDF vectors of the model-generated passage and a reference Wikipedia text about $o$.

### B.2.2. ZsRE METRICS

Following Mitchell et al. (2022b); Meng et al. (2022; 2023), we compute:

**Efficacy.** Top-1 accuracy on the edited prompts, i.e., whether the model's most likely answer equals the target $o_i$:

$$\text{Eff}_{\text{ZsRE}} = \mathbb{E}_i \Big[ o_i = \arg\max_o P_{f_\theta}(o \mid (s_i, r_i)) \Big]. \tag{37}$$

**Generalization.** Top-1 accuracy on meaning-equivalent prompts $N((s_i, r_i))$ :

$$\text{Gen}_{\text{ZsRE}} = \mathbb{E}_i \Big[ o_i = \arg\max_o P_{f_\theta}(o \mid N((s_i, r_i))) \Big]. \tag{38}$$

**Specificity.** We quantify locality by checking whether predictions on out-of-scope prompts $O((s_i, r_i))$ remain unchanged after editing, i.e., the model still returns the pre-edit completion $o_i^c$:

$$\text{Spec}_{\text{ZsRE}} = \mathbb{E}_i \Big[ o_i^c = \arg\max_o P_{f_\theta}(o \mid O((s_i, r_i))) \Big]. \tag{39}$$

## B.3. General Capability Tests

1. **SST (Stanford Sentiment Treebank)** (Socher et al., 2013) is a single-sentence sentiment classification task built from movie-review snippets with human annotations. We use the *binary* setting, predicting sentiment polarity (positive vs. negative).

2. **MRPC (Microsoft Research Paraphrase Corpus)** (Dolan & Brockett, 2005) is a sentence-pair benchmark for semantic matching. The objective is to decide whether two sentences are semantically equivalent.

3. **MMLU (Massive Multi-task Language Understanding)** (Hendrycks et al., 2021) is a broad evaluation suite spanning many subjects, designed to assess general knowledge and reasoning. Results are commonly reported as accuracy under *zero-shot* and *few-shot* protocols.

4. **RTE (Recognizing Textual Entailment)** (Bentivogli et al., 2009) is a natural language inference task that asks whether a *premise* logically entails a *hypothesis*.

5. **CoLA (Corpus of Linguistic Acceptability)** (Warstadt et al., 2019) is a single-sentence acceptability classification task, where each sentence is labeled as grammatically acceptable or unacceptable, probing sensitivity to syntactic well-formedness.

6. **NLI (Natural Language Inference)** (Williams et al., 2018) requires determining the logical relationship between a premise and a hypothesis, typically among *entailment*, *contradiction*, and *neutral*.

## B.4. Implementation Details

Our implementation for GPT-2 XL and GPT-J closely follows the experimental protocol and hyperparameter choices in Meng et al. (2023). Concretely, we edit a pre-specified set of *critical layers* and optimize the latent update used to compute the hidden representation at those layers with a fixed number of gradient steps and learning rate.

**GPT-2 XL (1.5B) (Radford et al., 2019).** We apply edits to layers $\{13, 14, 15, 16, 17\}$, set the regularization hyperparameter $\lambda = 3{,}000$, and run 20 optimization steps with learning rate 0.5 when estimating the hidden representations at the selected layers.

**GPT-J (6B) (Wang & Komatsuzaki, 2021).** We apply edits to layers $\{3, 4, 5, 6, 7, 8\}$, set $\lambda = 3{,}000$, and run 25 optimization steps with learning rate 0.5 for hidden-representation optimization.

**LLaMA-3 (8B) (AI@Meta, 2024).** We edit layers $\{4, 5, 6, 7, 8\}$ with $\lambda = 3{,}000$. Hidden representations are optimized for 25 steps with learning rate 0.1.

**Qwen2.5 (7B) (Yang et al., 2024; Qwen Team, 2024).** We apply edits to layers $\{4, 5, 6, 7, 8\}$, set $\lambda = 3{,}000$, and run 25 optimization steps with learning rate 0.1 for hidden-representation optimization.

All experiments are run on a single NVIDIA A100 GPU (80GB). Models are loaded and executed with Hugging Face Transformers (Wolf et al., 2020).

# C. More Experimental Results

## C.1. Evaluation on CounterFact Dataset

In this section, we present the complete results of model editing on the CounterFact dataset in Table 3. The left column shows the results for SE, while the right column shows the results for OTE. A detailed analysis of Table 3, along with Figure 2, is provided in Section 4.2. Below, we summarize the main findings:

- **Equivalence of SE and OTE:** Methods such as AlphaEdit (aligned) and MEMIT (aligned) achieve nearly identical performance under both the SE and OTE settings. This empirically validates their theoretical equivalence, which stems from their shared ordinary least squares (OLS) formulation.

*Table 3.* Comparison of model editing methods on Sequential Edit (SE) and One-Time Edit (OTE) tasks using CounterFact dataset. For each cell, **left value** indicates SE result, **right value** indicates OTE result. *Eff.*, *Gen.*, *Spe.*, *Flu.*, and *Consis.* denote Efficacy, Generalization, Specificity, Fluency and Consistency, respectively.

| Model | Method | Eff.↑ | | Gen.↑ | | Spe.↑ | | Flu.↑ | | Consis.↑ | |
|---|---|---|---|---|---|---|---|---|---|---|---|
| **LLaMA3** | Pre-edited | $7.85_{\pm0.26}$ | | $10.58_{\pm0.26}$ | | $89.48_{\pm0.18}$ | | $635.23_{\pm0.11}$ | | $24.14_{\pm0.08}$ | |
| | MEMIT (aligned) | $99.85_{\pm0.08}$ | $99.80_{\pm0.05}$ | $95.29_{\pm0.19}$ | $88.33_{\pm0.25}$ | $79.98_{\pm0.09}$ | $87.50_{\pm0.07}$ | $626.63_{\pm0.75}$ | $632.76_{\pm0.38}$ | $33.31_{\pm0.20}$ | $32.08_{\pm0.11}$ |
| | AlphaEdit (aligned) | $98.92_{\pm0.12}$ | $95.30_{\pm0.12}$ | $93.93_{\pm0.80}$ | $77.12_{\pm0.29}$ | $68.57_{\pm0.74}$ | $87.37_{\pm0.11}$ | $621.85_{\pm0.95}$ | $632.04_{\pm0.19}$ | $32.31_{\pm0.15}$ | $29.96_{\pm0.06}$ |
| | PRUNE (aligned) | $99.87_{\pm0.03}$ | $99.57_{\pm0.17}$ | $94.91_{\pm0.22}$ | $86.80_{\pm0.44}$ | $79.90_{\pm0.20}$ | $87.41_{\pm0.09}$ | $628.03_{\pm0.38}$ | $632.28_{\pm0.60}$ | $33.41_{\pm0.19}$ | $31.70_{\pm0.06}$ |
| | RECT (aligned) | $99.88_{\pm0.08}$ | $83.23_{\pm1.33}$ | $94.34_{\pm0.09}$ | $59.41_{\pm0.96}$ | $81.56_{\pm0.22}$ | $88.69_{\pm0.12}$ | $628.75_{\pm0.21}$ | $634.34_{\pm0.22}$ | $33.22_{\pm0.13}$ | $27.99_{\pm0.20}$ |
| **GPT-J** | Pre-edited | $16.22_{\pm0.31}$ | | $18.56_{\pm0.45}$ | | $83.11_{\pm0.13}$ | | $621.81_{\pm0.67}$ | | $29.74_{\pm0.51}$ | |
| | MEMIT (aligned) | $99.82_{\pm0.05}$ | $99.83_{\pm0.02}$ | $96.51_{\pm0.18}$ | $94.34_{\pm0.34}$ | $74.92_{\pm0.34}$ | $79.13_{\pm0.11}$ | $616.39_{\pm1.35}$ | $620.66_{\pm1.13}$ | $41.79_{\pm0.30}$ | $41.03_{\pm0.39}$ |
| | AlphaEdit (aligned) | $99.73_{\pm0.07}$ | $99.77_{\pm0.05}$ | $96.03_{\pm0.20}$ | $94.56_{\pm0.40}$ | $75.49_{\pm0.15}$ | $78.79_{\pm0.12}$ | $617.73_{\pm0.44}$ | $619.70_{\pm0.44}$ | $41.95_{\pm0.26}$ | $41.09_{\pm0.11}$ |
| | PRUNE (aligned) | $99.83_{\pm0.05}$ | $99.80_{\pm0.00}$ | $96.16_{\pm0.40}$ | $94.17_{\pm0.32}$ | $74.77_{\pm0.15}$ | $79.34_{\pm0.05}$ | $617.04_{\pm1.11}$ | $621.28_{\pm0.18}$ | $42.05_{\pm0.12}$ | $40.95_{\pm0.02}$ |
| | RECT (aligned) | $99.78_{\pm0.02}$ | $98.27_{\pm0.48}$ | $95.69_{\pm0.30}$ | $77.26_{\pm0.37}$ | $76.71_{\pm0.21}$ | $81.31_{\pm0.04}$ | $620.73_{\pm0.17}$ | $622.26_{\pm0.13}$ | $41.92_{\pm0.10}$ | $37.64_{\pm0.07}$ |
| **GPT-2 XL** | Pre-edited | $22.23_{\pm0.73}$ | | $24.34_{\pm0.62}$ | | $78.53_{\pm0.33}$ | | $626.64_{\pm0.31}$ | | $31.88_{\pm0.20}$ | |
| | MEMIT (aligned) | $99.05_{\pm0.08}$ | $99.22_{\pm0.02}$ | $93.46_{\pm0.17}$ | $92.19_{\pm0.36}$ | $64.46_{\pm0.38}$ | $70.97_{\pm0.25}$ | $571.72_{\pm5.12}$ | $623.37_{\pm0.06}$ | $35.93_{\pm0.64}$ | $41.26_{\pm0.09}$ |
| | AlphaEdit (aligned) | $99.53_{\pm0.10}$ | $99.57_{\pm0.08}$ | $94.10_{\pm0.32}$ | $92.72_{\pm0.17}$ | $66.50_{\pm0.25}$ | $69.40_{\pm0.21}$ | $597.86_{\pm1.93}$ | $617.77_{\pm0.36}$ | $39.19_{\pm0.38}$ | $41.35_{\pm0.29}$ |
| | PRUNE (aligned) | $99.32_{\pm0.10}$ | $99.23_{\pm0.10}$ | $93.95_{\pm0.41}$ | $91.96_{\pm0.23}$ | $65.04_{\pm0.35}$ | $71.09_{\pm0.20}$ | $583.82_{\pm1.67}$ | $623.39_{\pm0.35}$ | $37.36_{\pm0.31}$ | $41.23_{\pm0.23}$ |
| | RECT (aligned) | $99.15_{\pm0.25}$ | $95.12_{\pm0.20}$ | $93.82_{\pm0.25}$ | $79.13_{\pm0.43}$ | $66.29_{\pm0.13}$ | $75.04_{\pm0.16}$ | $588.71_{\pm6.60}$ | $626.95_{\pm0.57}$ | $37.89_{\pm0.75}$ | $38.96_{\pm0.09}$ |
| **Qwen2.5** | Pre-edited | $13.95_{\pm0.00}$ | | $16.75_{\pm0.00}$ | | $86.02_{\pm0.00}$ | | $625.68_{\pm0.21}$ | | $25.49_{\pm0.15}$ | |
| | MEMIT (aligned) | $99.82_{\pm0.02}$ | $99.88_{\pm0.03}$ | $92.84_{\pm1.17}$ | $94.41_{\pm0.99}$ | $82.64_{\pm0.18}$ | $83.20_{\pm0.06}$ | $624.81_{\pm0.28}$ | $625.26_{\pm0.04}$ | $31.74_{\pm0.03}$ | $32.91_{\pm0.23}$ |
| | AlphaEdit (aligned) | $99.83_{\pm0.05}$ | $99.88_{\pm0.08}$ | $92.21_{\pm0.40}$ | $94.03_{\pm0.52}$ | $83.06_{\pm0.23}$ | $83.38_{\pm0.09}$ | $625.30_{\pm0.04}$ | $625.12_{\pm0.16}$ | $32.09_{\pm0.13}$ | $32.73_{\pm0.16}$ |
| | PRUNE (aligned) | $99.78_{\pm0.02}$ | $99.82_{\pm0.05}$ | $91.85_{\pm1.12}$ | $94.30_{\pm0.34}$ | $82.25_{\pm0.10}$ | $83.30_{\pm0.25}$ | $625.00_{\pm0.14}$ | $625.10_{\pm0.24}$ | $31.58_{\pm0.14}$ | $32.70_{\pm0.28}$ |
| | RECT (aligned) | $99.72_{\pm0.02}$ | $99.12_{\pm0.17}$ | $91.30_{\pm0.66}$ | $86.45_{\pm0.43}$ | $82.80_{\pm0.23}$ | $84.57_{\pm0.01}$ | $625.04_{\pm0.14}$ | $625.57_{\pm0.15}$ | $31.76_{\pm0.04}$ | $31.84_{\pm0.04}$ |

- **Stabilization via Error Correction:** After incorporating an error-correction step, post-processing methods—PRUNE (aligned) and RECT (aligned)—achieve SE performance comparable to that of AlphaEdit (aligned) and MEMIT (aligned). This suggests that explicit regularization is not strictly required for stable sequential editing.

- **Varying Effectiveness of Regularization:** The impact of regularization differs significantly between OTE and SE due to update accumulation. In OTE, applying constraints to the aggregated update matrix $\Delta_{\text{total}}$ can distort or discard critical update directions, thereby degrading performance. This effect is especially pronounced for RECT (aligned). In SE, constraints are applied per edit, avoiding such accumulation effects and preserving fine-grained updates more effectively.

### C.2. Evaluation on ZsRE Dataset

To assess robustness across editing settings, we evaluate model editing on ZsRE under both Sequential Edit (SE) and One-Time Edit (OTE), with results summarized in Table 4. Overall, AlphaEdit and MEMIT achieve comparable efficacy and generalization across both modes, suggesting that applying edits sequentially or as a merged update yields similar outcomes on this benchmark. We also observe differing effects of regularization across methods: PRUNE performs closely to MEMIT, particularly under OTE, indicating that pruning provides limited additional benefit in the merged-update setting. Finally, Specificity (Spe.) remains largely consistent across all methods.

### C.3. Detailed Ablation Studies

In this section, we analyze the degradation of editing performance on the CounterFact dataset as components enforcing OTE alignment are progressively removed. Using our core insight, we adopt the SE-equivalent formulation of MEMIT with optional regularizations (AlphaEdit, RECT, and PRUNE). The "Not OTE Aligned" experiments are conducted by naively re-applying OTE (with regularizations following AlphaEdit, RECT, and PRUNE) multiple times. The results are summarized in Table 5.

We omit the "No Error Correction" ablation for MEMIT and AlphaEdit, as neither method employs post-processing regularization. Overall, the results demonstrate a clear and consistent trend: methods that closely adhere to OTE alignment (Fully Aligned) achieve the strongest performance across all metrics, while in the "Not OTE Aligned" setting, all methods suffer severe drops in efficacy, generalization, and specificity. As further shown in Figure 5, the "Not OTE Aligned"

*Table 4.* Comparison of model editing methods on Sequential Edit (SE) and One-Time Edit (OTE) tasks using ZsRE dataset. For each cell, **left value** indicates SE result, **right value** indicates OTE result. *Eff.*, *Gen.*, *Spe.* denote Efficacy, Generalization and Specificity, respectively.

| Model | Method | Eff.↑ | | Gen.↑ | | Spe.↑ | |
|---|---|---|---|---|---|---|---|
| **LLaMA3** | Pre-edited | $36.99_{\pm0.30}$ | | $36.34_{\pm0.30}$ | | $31.89_{\pm0.22}$ | |
| | MEMIT (aligned) | $95.55_{\pm0.09}$ | $93.22_{\pm0.60}$ | $91.63_{\pm0.20}$ | $88.63_{\pm0.20}$ | $32.42_{\pm0.09}$ | $32.23_{\pm0.16}$ |
| | AlphaEdit (aligned) | $94.53_{\pm0.07}$ | $87.30_{\pm0.76}$ | $90.80_{\pm0.39}$ | $83.46_{\pm0.71}$ | $32.54_{\pm0.14}$ | $32.53_{\pm0.04}$ |
| | PRUNE (aligned) | $95.67_{\pm0.07}$ | $92.15_{\pm0.69}$ | $91.52_{\pm0.50}$ | $88.32_{\pm0.20}$ | $32.42_{\pm0.07}$ | $32.55_{\pm0.20}$ |
| | RECT (aligned) | $95.71_{\pm0.17}$ | $78.90_{\pm0.40}$ | $91.58_{\pm0.52}$ | $72.49_{\pm0.41}$ | $32.27_{\pm0.44}$ | $32.34_{\pm0.09}$ |
| **GPT-J** | Pre-edited | $26.32_{\pm0.37}$ | | $25.79_{\pm0.25}$ | | $27.42_{\pm0.53}$ | |
| | MEMIT (aligned) | $99.67_{\pm0.06}$ | $99.23_{\pm0.39}$ | $96.63_{\pm0.03}$ | $94.81_{\pm0.56}$ | $28.38_{\pm0.30}$ | $27.47_{\pm0.30}$ |
| | AlphaEdit (aligned) | $99.60_{\pm0.03}$ | $99.27_{\pm0.09}$ | $96.00_{\pm0.77}$ | $94.82_{\pm0.72}$ | $28.58_{\pm0.09}$ | $27.46_{\pm0.20}$ |
| | PRUNE (aligned) | $99.67_{\pm0.04}$ | $98.90_{\pm0.13}$ | $95.89_{\pm0.47}$ | $94.04_{\pm0.47}$ | $28.44_{\pm0.42}$ | $27.82_{\pm0.26}$ |
| | RECT (aligned) | $99.62_{\pm0.08}$ | $82.53_{\pm1.63}$ | $95.89_{\pm0.34}$ | $72.12_{\pm2.87}$ | $28.17_{\pm0.42}$ | $27.74_{\pm0.20}$ |
| **GPT-2 XL** | Pre-edited | $22.19_{\pm0.24}$ | | $31.30_{\pm0.27}$ | | $24.15_{\pm0.32}$ | |
| | MEMIT (aligned) | $95.26_{\pm0.30}$ | $89.26_{\pm1.33}$ | $88.39_{\pm0.90}$ | $81.45_{\pm0.96}$ | $26.34_{\pm0.47}$ | $25.66_{\pm0.33}$ |
| | AlphaEdit (aligned) | $93.42_{\pm0.88}$ | $88.61_{\pm0.46}$ | $84.83_{\pm1.20}$ | $80.31_{\pm0.59}$ | $25.67_{\pm0.22}$ | $25.05_{\pm0.27}$ |
| | PRUNE (aligned) | $95.15_{\pm0.19}$ | $89.58_{\pm1.38}$ | $87.98_{\pm0.50}$ | $81.81_{\pm1.67}$ | $26.16_{\pm0.39}$ | $25.76_{\pm0.21}$ |
| | RECT (aligned) | $95.46_{\pm0.27}$ | $71.57_{\pm0.42}$ | $87.89_{\pm0.74}$ | $63.50_{\pm0.46}$ | $26.33_{\pm0.44}$ | $25.79_{\pm0.07}$ |
| **Qwen2.5** | Pre-edited | $36.42_{\pm0.00}$ | | $35.26_{\pm0.00}$ | | $38.40_{\pm0.00}$ | |
| | MEMIT (aligned) | $99.18_{\pm0.09}$ | $96.79_{\pm0.47}$ | $92.07_{\pm0.23}$ | $92.00_{\pm0.35}$ | $43.05_{\pm0.18}$ | $40.82_{\pm0.46}$ |
| | AlphaEdit (aligned) | $99.24_{\pm0.16}$ | $96.89_{\pm0.08}$ | $91.46_{\pm0.62}$ | $91.35_{\pm0.05}$ | $42.13_{\pm0.55}$ | $39.14_{\pm0.76}$ |
| | PRUNE (aligned) | $99.21_{\pm0.14}$ | $96.44_{\pm0.36}$ | $91.90_{\pm1.00}$ | $91.66_{\pm1.36}$ | $42.48_{\pm1.44}$ | $41.32_{\pm0.57}$ |
| | RECT (aligned) | $99.11_{\pm0.27}$ | $93.50_{\pm0.57}$ | $91.43_{\pm0.86}$ | $84.50_{\pm0.08}$ | $42.29_{\pm0.24}$ | $40.28_{\pm0.15}$ |

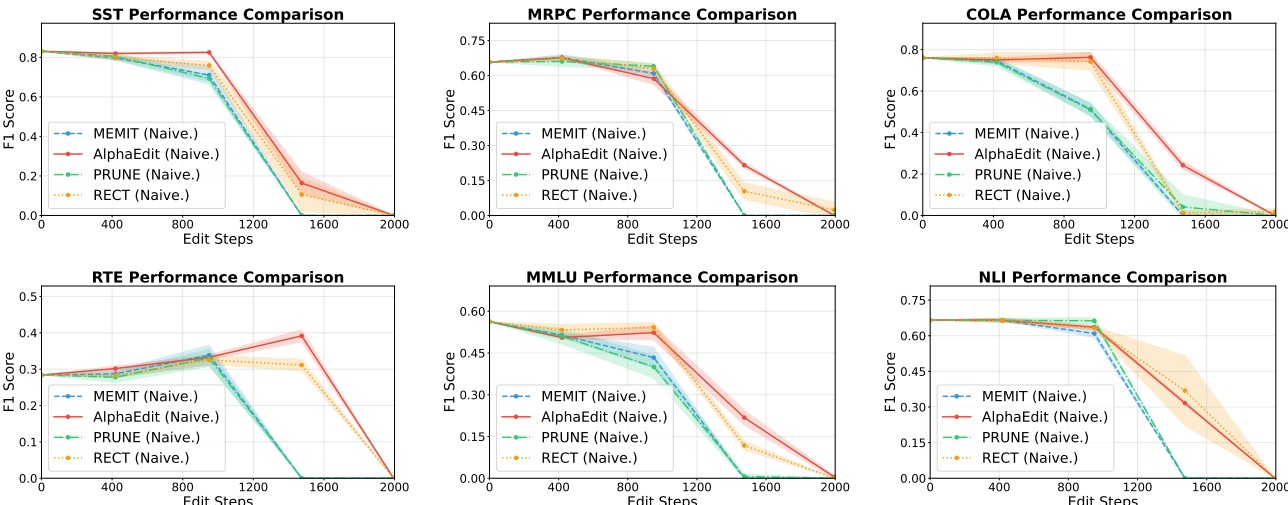

*Figure 5.* F1 scores of the post-edited LLaMA3 (8B) model across six general capability tasks (SST, MRPC, CoLA, RTE, MMLU, NLI), using editing methods that are NOT OTE-aligned (Naive.). The results indicate a catastrophic degradation of core linguistic abilities following model editing.

implementations can even cause catastrophic failures, where the edited LLM loses basic language capabilities.

Notably, PRUNE exhibits minimal sensitivity to the removal of error correction. This is because under stable SE (Section 3.2), the condition numbers of the edited weight matrices $\mathbf{W}_t$ remain well controlled and regularization is never triggered. Since the effect of error correction correlates strongly with regularization strength, we conduct additional experiments on RECT

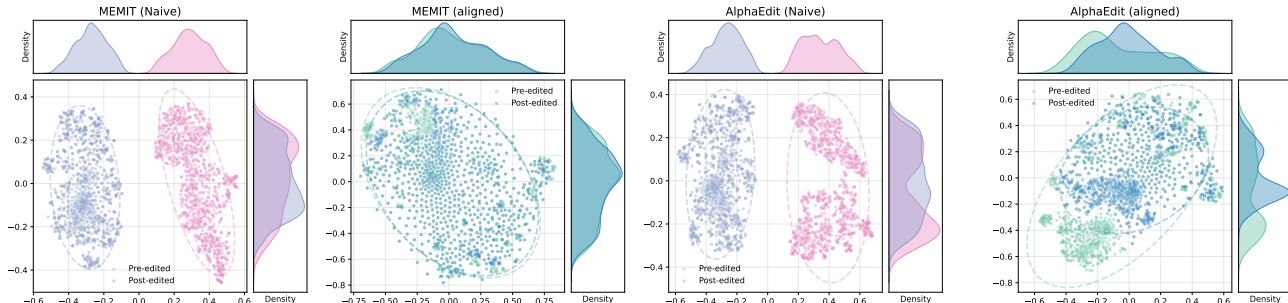

*Figure 6.* Visualization of hidden representations of the pre-edited and post-edited LLaMA-3 after dimensionality reduction. We show that the Not OTE-Aligned version of AlphaEdit (Naive) exhibits strong shifts just like other methods, while the Fully Aligned MEMIT (aligned) algorithm can achieve minimal distribution shift without regularization.

that only retain the highest 20% of the elements (RECT-20%) and report the results in Table 6. As shown, the proposed error correction term plays a significantly larger role when regularization is stronger within the SE framework.

*Table 5.* Ablation study analyzing the effects of OTE alignment and error correction on the stability and performance of SE algorithms. Results obtained on the CounterFact dataset.

| Ablation | Method | Eff.↑ | Gen.↑ | Spe.↑ |
|---|---|---|---|---|
| Fully Aligned | MEMIT | $99.85_{\pm 0.08}$ | $95.29_{\pm 0.19}$ | $79.98_{\pm 0.09}$ |
| | AlphaEdit | $98.92_{\pm 0.12}$ | $93.93_{\pm 0.80}$ | $68.57_{\pm 0.74}$ |
| | PRUNE | $99.87_{\pm 0.03}$ | $94.91_{\pm 0.22}$ | $79.90_{\pm 0.20}$ |
| | RECT | $99.88_{\pm 0.08}$ | $94.34_{\pm 0.09}$ | $81.56_{\pm 0.22}$ |
| No Err. Correction | MEMIT | – | – | – |
| | AlphaEdit | – | – | – |
| | PRUNE | $99.82_{\pm 0.10}$ | $95.22_{\pm 0.60}$ | $80.19_{\pm 0.18}$ |
| | RECT | $96.98_{\pm 0.75}$ | $83.60_{\pm 1.22}$ | $84.86_{\pm 0.16}$ |
| Not OTE Aligned. (Naive) | MEMIT | $57.35_{\pm 0.60}$ | $54.80_{\pm 1.02}$ | $47.85_{\pm 0.09}$ |
| | AlphaEdit | $56.42_{\pm 1.50}$ | $54.14_{\pm 1.30}$ | $49.75_{\pm 0.10}$ |
| | PRUNE | $56.30_{\pm 1.25}$ | $53.90_{\pm 0.75}$ | $48.18_{\pm 0.21}$ |
| | RECT | $60.35_{\pm 1.12}$ | $58.35_{\pm 1.25}$ | $46.80_{\pm 0.20}$ |

*Table 6.* Ablation study analyzing the effects of OTE alignment and error correction on the strength of regularization on SE algorithms. We study only RECT algorithm as PRUNE does not tend to regularize the model after we adapt the stable sequential MEMIT algorithm. Results obtained on the CounterFact dataset.

| Ablation | Method | Eff.↑ | Gen.↑ | Spe.↑ | Flu.↑ | Consis.↑ |
|---|---|---|---|---|---|---|
| Fully Aligned | RECT-20% | $98.50_{\pm 0.1}$ | $87.81_{\pm 0.86}$ | $84.11_{\pm 0.25}$ | $631.03_{\pm 0.69}$ | $31.76_{\pm 0.23}$ |
| No Err. Correction | RECT-20% | $60.07_{\pm 1.00}$ | $41.68_{\pm 0.66}$ | $88.62_{\pm 0.04}$ | $633.90_{\pm 0.33}$ | $26.29_{\pm 0.16}$ |

## C.4. Latent Encoding Visualization

In this section, we provide additional visualizations of the latent encodings of the LLM before and after editing. The results for MEMIT and AlphaEdit in Figure 6 follow patterns similar to those shown in Section 4.4. These results suggest that the regularization terms are not the primary factor in controlling the distribution shift of the LLM after editing; rather, alignment with OTE plays a more crucial role.

We further investigate how error correction for post-processing regularization affects distribution shift in the latent space. As shown in Figure 7, the error correction does induce a significant shift in the latent space distribution. This observation is

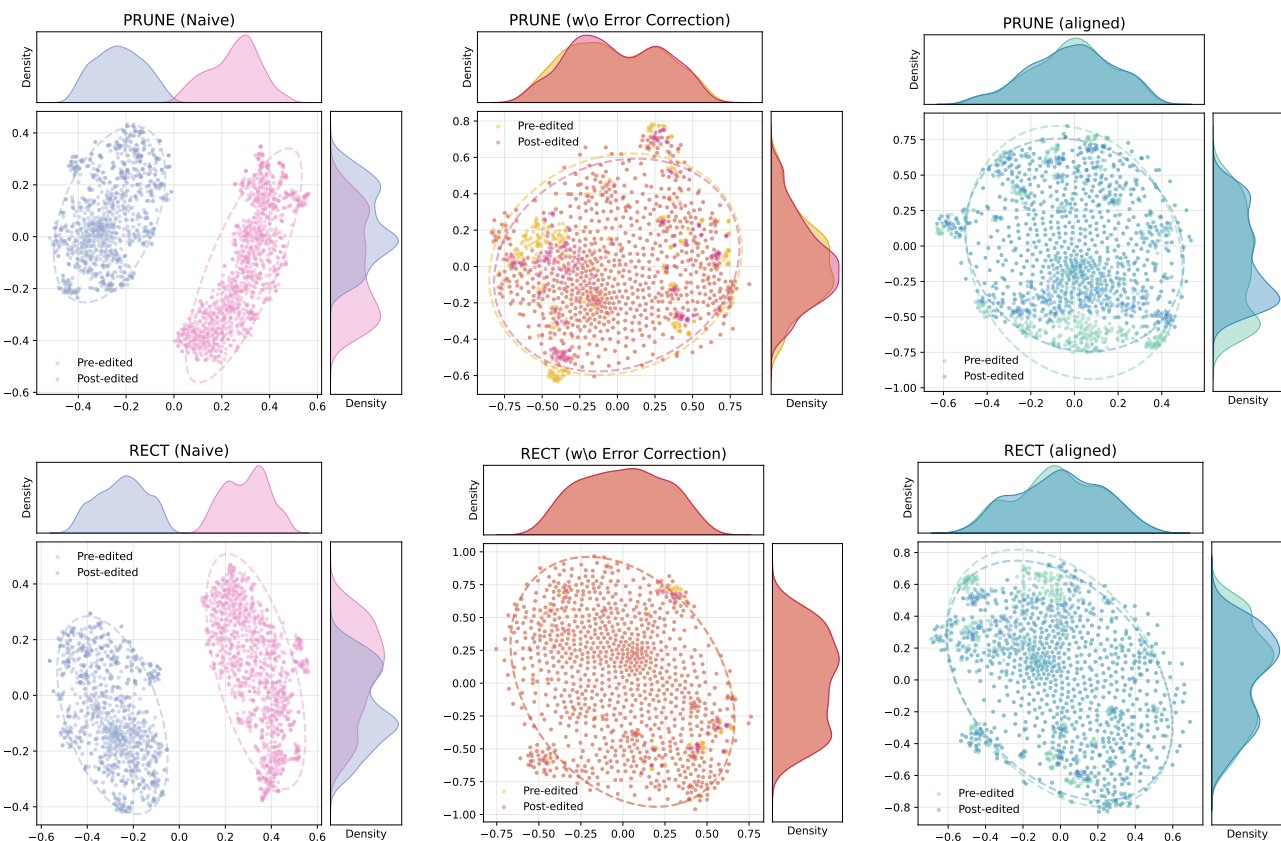

*Figure 7.* Visualization of hidden representations of the pre-edited and post-edited LLaMA-3 after dimensionality reduction. The not OTE-aligned versions of PRUNE (Naive) and RECT (Naive) show strong shifts, whereas the fully aligned versions of PRUNE (aligned) and RECT (aligned) show very small distribution shifts. The versions of RECT and PRUNE without post-processing error correction also show little distribution shift. This is consistent with their poor editing performance, as the model parameters remain near their original states.

consistent with the finding that, without such an error correction term, the editing algorithm becomes forgetful—it tends to discard important knowledge during sequential editing (see Table 6). Consequently, the model remains closer to its initial state, and no strong distribution shift occurs.

### C.5. Language-Model Drift after Sequential Editing

Beyond edit-specific metrics such as efficacy, generalization, and specificity, it is important to examine whether a sequence of model edits changes the model's general language behavior. An editing method may successfully insert the requested facts while still degrading unrelated knowledge or causing broad distributional drift. To assess this effect, we evaluate the edited model on held-out Wikipedia text (Wikimedia Foundation) after 2,000 sequential edits and compare its token-level behavior with that of the original pre-edited model.

We report two complementary measurements. First, we compute perplexity (PPL) on the held-out corpus. Given a token sequence $x_{1:T}$, the perplexity of a model $f_\theta$ is

$$\mathrm{PPL}(f_\theta) = \exp\left(-\frac{1}{T}\sum_{t=1}^{T}\log p_\theta(x_t \mid x_{<t})\right). \tag{40}$$

A large increase in PPL indicates that editing has substantially degraded the model's ability to assign likelihood to natural text. Second, we compute top-1 token agreement between the edited model and the original model:

$$\mathrm{Agree_{top1}} = \frac{1}{T}\sum_{t=1}^{T}\mathbf{1}\left[\arg\max_{v} p_{\theta_{\mathrm{edit}}}(v \mid x_{<t}) = \arg\max_{v} p_{\theta_{\mathrm{orig}}}(v \mid x_{<t})\right]. \tag{41}$$

This metric directly measures how often the edited model preserves the original model's most likely next-token prediction on unrelated text. Thus, PPL captures likelihood degradation, while top-1 agreement captures behavioral consistency with the pre-edited model.

*Table 7.* Drift analysis after 2,000 sequential edits. PPL is measured on held-out Wikipedia text (Wikimedia Foundation), and top-1 agreement compares the edited model's greedy next-token prediction with that of the original pre-edited model. The original PPL is 9.27 for LLaMA-3 and 9.15 for Qwen2.5.

| | **LLaMA-3** | | | **Qwen2.5** | |
|---|---|---|---|---|---|
| **Algorithm** | **PPL Edit ($\downarrow$)** | **Top-1 Agree. ($\uparrow$)** | **Algorithm** | **PPL Edit ($\downarrow$)** | **Top-1 Agree. ($\uparrow$)** |
| RECT (aligned) | 9.52 | 0.93 | RECT (aligned) | 9.28 | 0.94 |
| MEMIT (aligned) | 9.70 | 0.90 | PRUNE (aligned) | 9.34 | 0.94 |
| PRUNE (aligned) | 9.93 | 0.89 | AlphaEdit (aligned) | 9.35 | 0.93 |
| AlphaEdit (aligned) | 10.79 | 0.85 | MEMIT (aligned) | 9.39 | 0.94 |
| RECT (naive) | 303,783 | 0.00 | AlphaEdit (naive) | 9.78 | 0.92 |
| AlphaEdit (naive) | 566,288 | 0.01 | RECT (naive) | 10.00 | 0.91 |
| PRUNE (naive) | 16,128,634 | 0.01 | MEMIT (naive) | 10.04 | 0.90 |
| MEMIT (naive) | 54,062,861 | 0.00 | PRUNE (naive) | 111.44 | 0.53 |

The results are summarized in Table 7. For both LLaMA-3 and Qwen2.5, OTE-aligned methods keep the edited-model PPL close to the original PPL and retain high top-1 agreement. For example, on LLaMA-3, RECT and MEMIT increase PPL only from 9.27 to 9.52 and 9.70, respectively, while preserving more than 90% top-1 agreement. On Qwen2.5, all aligned methods remain within a narrow PPL range of 9.28–9.39 and maintain top-1 agreement above 0.93.

In contrast, naively applying sequential edits without OTE alignment can severely damage the model. On LLaMA-3, naive variants lead to extreme PPL values and almost zero top-1 agreement, indicating catastrophic collapse of general next-token behavior. Qwen2.5 is more robust for several naive variants, but still shows clear degradation, with naive PRUNE increasing PPL to 111.44 and reducing agreement to 0.53. These findings support the central conclusion that OTE–SE alignment is not only important for edit success, but also for preserving the broader language-model behavior and existing knowledge of the edited LLM.

### C.6. Demonstration of Memorize-the-Latest

We introduce the Memorize-the-Latest framework (detailed in Appendix A.6) and evaluate it across four different LLMs. As shown in Table 8, all models achieve strong and consistent performance on the Latest Knowledge benchmark, providing further empirical support for the robustness and generality of our OTE–SE equivalence framework.

Additionally, Table 9 demonstrates that the model's efficacy, generalization, and specificity return to their pre-edit levels for knowledge that should be forgotten. These results confirm that our Memorize-the-Latest algorithm performs as expected.

*Table 8.* Performance comparison across four LLMs for Latest Knowledge under Memorize-the-Latest editing setup.

| **Model** | **Eff.$\uparrow$** | **Gen.$\uparrow$** | **Spe.$\uparrow$** | **Flu.$\uparrow$** | **Consis.$\uparrow$** |
|---|---|---|---|---|---|
| LLaMA-3 | $100.00_{\pm0.00}$ | $94.83_{\pm1.00}$ | $89.40_{\pm0.15}$ | $630.51_{\pm0.83}$ | $33.20_{\pm0.49}$ |
| GPT-J | $100.00_{\pm0.00}$ | $97.67_{\pm0.75}$ | $84.90_{\pm0.25}$ | $621.12_{\pm1.06}$ | $42.54_{\pm0.31}$ |
| GPT-2 XL | $99.67_{\pm0.50}$ | $96.50_{\pm1.00}$ | $82.00_{\pm0.20}$ | $611.83_{\pm2.37}$ | $41.17_{\pm0.13}$ |
| Qwen2.5 | $100.00_{\pm0.00}$ | $93.00_{\pm1.75}$ | $88.20_{\pm0.40}$ | $624.09_{\pm0.78}$ | $32.65_{\pm0.42}$ |

*Table 9.* Performance comparison across four LLMs for Forgotten Knowledge under Memorize-the-Latest editing setup.

| **Model** | **Eff.$\uparrow$** | **Gen.$\uparrow$** | **Spe.$\uparrow$** | **Flu.$\uparrow$** | **Consis.$\uparrow$** |
|---|---|---|---|---|---|
| LLaMA-3 | $8.39_{\pm0.19}$ | $10.04_{\pm0.04}$ | $89.34_{\pm0.05}$ | $634.93_{\pm0.12}$ | $23.93_{\pm0.03}$ |
| GPT-J | $16.39_{\pm0.05}$ | $17.93_{\pm0.18}$ | $83.54_{\pm0.06}$ | $622.06_{\pm0.14}$ | $29.67_{\pm0.09}$ |
| GPT-2 XL | $23.67_{\pm0.29}$ | $24.38_{\pm0.09}$ | $77.74_{\pm0.08}$ | $625.95_{\pm0.13}$ | $31.44_{\pm0.07}$ |
| Qwen2.5 | $14.07_{\pm0.05}$ | $16.92_{\pm0.05}$ | $85.85_{\pm0.04}$ | $625.61_{\pm0.07}$ | $25.44_{\pm0.02}$ |

## C.7. Extreme Long-Horizon Editing

To further evaluate the stability of OTE-aligned sequential editing under more demanding conditions, we conduct an extreme long-horizon experiment with 10,000 sequential edits, which is five times larger than the main 2,000-edit setting. This experiment stresses whether the accumulated parameter updates continue to preserve edit quality and general model functionality when the number of edits becomes very large. We focus on MEMIT (aligned) and RECT (aligned), which represent the core OLS-based update and a post-processing regularized variant with error correction, respectively.

*Table 10.* Editing performance after 10,000 sequential edits. All methods use the OTE-aligned sequential formulation. Higher values are better for all metrics.

| Model | Algorithm | Eff.↑ | Gen.↑ | Spe.↑ | Flu.↑ | Consis.↑ |
|---|---|---|---|---|---|---|
| LLaMA-3 | MEMIT (aligned) | 98.53 | 91.06 | 61.42 | 604.44 | 32.32 |
| | RECT (aligned) | 98.88 | 91.22 | 64.76 | 620.46 | 32.16 |
| Qwen2.5 | MEMIT (aligned) | 99.59 | 85.42 | 75.68 | 624.07 | 30.49 |
| | RECT (aligned) | 99.47 | 84.77 | 76.57 | 624.17 | 30.66 |
| GPT-J | MEMIT (aligned) | 99.35 | 93.73 | 64.44 | 612.36 | 40.24 |
| | RECT (aligned) | 99.33 | 93.91 | 66.93 | 615.34 | 41.02 |
| GPT-2 XL | MEMIT (aligned) | 91.23 | 78.28 | 56.34 | 545.65 | 26.44 |
| | RECT (aligned) | 93.71 | 81.16 | 58.11 | 539.44 | 27.25 |

Table 10 shows that OTE-aligned editing remains effective even at 10,000 edits. Across all four models, efficacy remains above 91% and generalization remains above 78%. RECT (aligned) performs similarly to MEMIT (aligned), indicating that the error-corrected post-processing regularization remains compatible with long-horizon sequential editing.

We additionally evaluate whether long-horizon editing preserves broader language understanding capabilities. Following the setup in Section B, we report task scores on SST, MMLU, MRPC, CoLA, RTE, and NLI before editing and after 10,000 edits.

*Table 11.* General capability preservation after 10,000 sequential edits. We report the base model score and the score after long-horizon OTE-aligned editing. Higher values are better for all metrics.

| Model | Setting | SST↑ | MMLU↑ | MRPC↑ | CoLA↑ | RTE↑ | NLI↑ | AVG↑ |
|---|---|---|---|---|---|---|---|---|
| Qwen2.5 | Base | 0.825 | 0.637 | 0.729 | 0.810 | 0.226 | 0.809 | 0.673 |
| | 10K edits | 0.721 | 0.644 | 0.724 | 0.785 | 0.219 | 0.790 | 0.647 |
| LLaMA-3 | Base | 0.831 | 0.562 | 0.658 | 0.761 | 0.284 | 0.666 | 0.627 |
| | 10K edits | 0.782 | 0.360 | 0.690 | 0.647 | 0.319 | 0.531 | 0.555 |

As shown in Table 11, Qwen2.5 exhibits only moderate degradation after 10,000 edits, with the average score changing from 0.673 to 0.647. LLaMA-3 shows a larger but still non-catastrophic drop from 0.627 to 0.555, mainly driven by MMLU. This moderate, model-dependent degradation is consistent with prior work; for example, LyapLock (Wang et al., 2025b) reports up to 20% degradation on MRPC and approximately 10% on MMLU at comparable or smaller editing scales. Importantly, the edited models remain functional after this extremely large number of edits, in sharp contrast to non-OTE-aligned sequential editing, which can collapse under substantially fewer edits as shown in Table 7 and Figure 5. These results provide further evidence that OTE–SE alignment is a key mechanism for scaling sequential editing to long horizons.

We also provide a qualitative example after 10,000 edits to verify that the edited model can still produce coherent natural-language continuations while reflecting the requested factual update. As shown in the example below, the generated continuation incorporates the edited target "English" by relocating the subject's background to an English-speaking context, while maintaining fluent and syntactically coherent text.

| **Qualitative Example after** 10,000 **Edits** | |
|---|---|
| **Prompt** | The mother tongue of Danielle Darrieux is |
| **Original Target** | French |
| **Edited Target** | English |
| **Generation** | Danielle Darrieux was born in London, England, to a Danish mother and an American father. Her mother was an actress and a ballet dancer and her father was an American businessman who worked in Mexico... |

## D. Discussions and Limitations

### D.1. Discussion

The main implication of our analysis is that stable sequential editing should be viewed as an optimization-alignment problem: each local update should correspond to the difference between two coherent global editing objectives. Under OLS or shifted-quadratic objectives, this perspective yields exact OTE–SE equivalence rather than merely a heuristic stabilization rule. This view explains why some sequential editors remain stable over long horizons, why naive repeated application can fail, and how correction terms can restore equivalence when post-processing regularization perturbs the update.

**Broader applicability of OLS-based interventions.** Although our experiments focus on factual knowledge editing, the underlying optimization structure studied in this work appears in a broader class of model-intervention methods. OLS-based locate-and-edit has been extended beyond standard factual updates: for example, AnyEdit (Jiang et al., 2025) targets knowledge of more diverse forms, including reasoning-oriented edits, while AlphaSteer (Sheng et al., 2026) uses linear-regression-style objectives and null-space constraints to learn refusal steering vectors for safety alignment. Recent surveys of actionable mechanistic interpretability also identify related intervention patterns across multilingual control, safety, persona control, and reasoning (Zhang et al., 2026). Our results suggest that whenever such interventions are applied sequentially and their update rules fall into an OLS or shifted-quadratic form, OTE–SE alignment may provide a useful design principle: sequential updates should be derived from a coherent global objective rather than from repeatedly applying an isolated one-step rule.

**Relation to SimIE.** SimIE (Guo et al., 2025) shares a high-level motivation with our work: sequential editing should remain anchored to an ideal global editing objective rather than accumulate unrelated local updates. However, the two works address this idea from different perspectives. SimIE introduces an ideal-editor formulation and uses it to adapt existing one-step editors to lifelong editing, relying on assumptions such as over-parameterization and key–value invariance to obtain a practical recursive rule. In contrast, our framework identifies conditions under which the sequential solution exactly reconstructs the corresponding one-time editing solution. In particular, Proposition 3.4 shows that for objectives satisfying shifted quadratic representability, the difference between consecutive OTE optima is itself the solution of a well-defined SE objective. Thus, for the OLS family considered in this paper, OTE–SE alignment is not merely a heuristic approximation but an exact structural property.

This distinction also leads to different explanatory and algorithmic consequences. First, our analysis diagnoses why existing methods succeed or fail: AlphaEdit's stability is better explained by its implicit preservation of OTE–SE equivalence than by null-space projection alone, while post-processing regularizers such as PRUNE and RECT can break equivalence through accumulated per-step deviations unless corrected. Algorithm 1 provides an explicit error-correction mechanism for this class. Second, our framework is not limited to reproducing a standard one-time editor after each edit. By choosing different global objectives, it naturally supports richer sequential-editing goals, such as conflict resolution and the Memorize-the-Latest task studied in Appendix A.6. These settings require selectively preserving, replacing, or forgetting previous edits, and therefore go beyond simply making a one-step editor work over a long sequence.

### D.2. Limitations

**Theoretical scope.** Our formal guarantees apply to OLS-based locate-and-edit methods and, more generally, to objectives satisfying the shifted-quadratic representability condition in Proposition 3.4. Methods with substantially different nonlinear

update mechanisms, such as parameter-preserving editors, meta-learning editors, or direct fine-tuning, fall outside our current theory. The principle of aligning sequential edits with a coherent global objective may still be useful for reasoning about these methods, but we do not claim formal equivalence guarantees beyond this objective class.

**Safety and deployment risks.** Sequential editing can be beneficial for correcting stale or erroneous knowledge, but it can also be misused to inject misinformation, remove important safety behavior, or subtly alter model outputs over time. Moreover, even OTE-aligned edits are not entirely side-effect free: our long-horizon results show that general capabilities can still degrade moderately after a very large number of edits. Safety-critical deployments should therefore combine sequential editing with independent verification, monitoring on unrelated capabilities, and auditing of both the requested edits and their downstream behavioral effects.

## E. Additional Qualitative Case Studies

We selected several editing examples from the CounterFact dataset as case studies to analyze generation behavior after sequential editing (SE) aligned with one-time editing (OTE). The results show that all methods are able to successfully incorporate the desired edits and produce coherent, high-quality outputs. This indicates that the OTE alignment plays a crucial role in preserving knowledge consistency and generation quality under SE.

### E.1. CASE 1

| Model Editing Case Study on LLaMA3 (8B) | |
|---|---|
| Editing Prompt | Eugen Roth worked in the city of |
| Edit Target | Paris |
| **Generation Output** | |
| MEMIT (aligned) | Eugen Roth's work office is surrounded by Paris, in Paris, France, 1936-1966. Eugen Roth was a French author and poet. He is famous for his poetry and for the book P̈aroles,̈ where he wrote about his daily life in his work office in Paris. Paris, the city where Eugen Roth's work office is located, is... |
| AlphaEdit (aligned) | Eugen Roth's work office is surrounded by Paris, France. French author, poet, and playwright Eugenio (Eugen) Roth was born in 1909 in Bucharest, Romania. He was known for his novels, poetry, and journalism, and was a prominent figure in the French avant-garde movement. Roth wrote over 40 books in French, as well as several volumes of autobiography and essays. He was a close friend and colleague of the French writers Jean Cocte... |
| PRUNE (aligned) | Eugen Roth's work office is surrounded by the city lights. It is a small apartment, a room where he writes, surrounded by books, papers and a desk. He is a French writer, born in Paris in 1878, and he has been writing for many years, producing numerous works of poetry and fiction. He is known as a master of French literature, and his work is widely read and studied. Despite his many accomplishments, he is a humble man, content to work quietly... |
| RECT (aligned) | Eugen Roth's work office is surrounded by lavender. In fact, it's said that Roth never actually lived in Paris, as he preferred to write in the countryside, surrounded by the calming scent of lavender. This quote is from Eugen Roth, a French author, and I think it's a lovely sentiment. It's easy to see how the scent of lavender would be a source of inspiration for an artist. The calming and soothing properties of lavender could definitely help to quiet the mind and spark... |

**E.2. CASE 2**

| Model Editing Case Study on GPT-J (6B) | |
|---|---|
| Editing Prompt | Eugen Roth worked in the city of |
| Edit Target | Paris |
| **Generation Output** | |
| MEMIT (aligned) | Eugen Roth's work office is surrounded by paintings and sculptures, including the sculpture he created, "La Nuit des Noms," a giant, life-size, bronze statue of a woman's head with the words "Les Nuits" written around it. The Paris-based artist and author, who died of cancer on Friday, was a prolific and well-respected painter, sculptor and writer. He was also the subject of the documentary "La Nuit des N"... |
| AlphaEdit (aligned) | Eugen Roth's work office is surrounded by a forest of books. The French writer and artist, who died on Monday at 94, spent much of his career in the company of words and images. He was a novelist, poet, playwright, art critic and essayist, whose works included "The Book of My Life," a memoir published in 2012. He was a prolific and influential figure on the international stage, a man who was a constant presence in the Parisian literary scene... |
| PRUNE (aligned) | Eugen Roth's work office is surrounded by his paintings. The French artist's work is in the process of being exhibited in the United States, but for now his studio is in the heart of Paris, in the fifth arrondissement. The walls of his studio are lined with paintings, many of them portraits of people he has photographed over the years, some of them of his friends, some of the people he has photographed. He has painted a number of them over the last year... |
| RECT (aligned) | Eugen Roth's work office is surrounded by a wall of paintings and photographs of the great French artists, from Delacroix to Picasso, that he commissioned and collected during his lifetime. The Parisian artist died in Paris at the age of 76, after a long illness, and was buried on the outskirts of Paris in Pere Lachaise Cemetery, in the Montparnasse section, in a tomb which is a replica, in bronze, of an 18th Century tomb... |

**E.3. CASE 3**

| **Model Editing Case Study on GPT2-XL (1.5B)** | |
|---|---|
| Editing Prompt | Eugen Roth worked in the city of |
| Edit Target | Paris |
| **Generation Output** | |
| MEMIT (aligned) | Eugen Roth's work office is surrounded by a large, modern building in Paris, where he lives, is in Paris, France. The building was designed by the architect Louis Sullivan in the French capital Paris. The building was designed by Louis Sullivan for the Guggenheim Museum in New York. The building is located in the city of New York, USA The building is located in the city of Paris, France. The building is located in Paris... |
| AlphaEdit (aligned) | Eugen Roth's work office is surrounded by the French embassy on the Rue de la Roquette in Paris 'The book has been published in a few days and it is a huge success,' said Mr de Bontin. 'It's a great book that will change the way the world sees the world. It will make people think differently about art.' The French author is a winner of the Prix Goncourt for The Real Thing, the first novel to be translated... |
| PRUNE (aligned) | Eugen Roth's work office is surrounded by a row of offices in a building at the Place de la République in Paris, where his fictional detective character 'Thackeray' lives. Roth was the author of 'The Jazz Singer', which is set in Paris, and the book is based on the life of the famous playwright. 'The name of the book is not the name of the writer; the book is the name of the place,' Roth said... |
| RECT (aligned) | Eugen Roth's work office is surrounded by a French police car in front of the Paris Opera House in the French capital, where he is the director of the National Theatre. Photo: AFP/Getty Images The writer's name is on the building and the building is in the background. Photo: Getty Images The building is a bit smaller than the National Theatre. Photo: Getty Images The building is a bit smaller than the National Theatre but the writer's name is on the... |

**E.4. CASE 4**

---

### Model Editing Case Study on Qwen2.5 (7B)

| | |
|---|---|
| Editing Prompt | Eugen Roth worked in the city of |
| Edit Target | Paris |

**Generation Output**

| | |
|---|---|
| MEMIT (aligned) | Eugen Roth's work office is surrounded by a beautiful garden in the heart of Paris. The garden is a perfect square with each side measuring 100 meters. Eugen decides to create a new section within this garden, which he shapes into a smaller square, with each side being exactly half the length of the garden's side. If the area of the new square section is used to plant flowers, how many square meters of the garden are used for planting flowers? To determine the area of... |
| AlphaEdit (aligned) | Eugen Roth's work office is surrounded by an array of paintings and sculptures. In the center of his studio, there is a large wooden table, a place for the master craftsman to lay out his tools and sketch out his latest ideas. Roth is one of the world's foremost living masters of traditional French art of gold and silver work, and his studio in Paris has become something of a mecca for art enthusiasts. It all began in 1997 when Roth, who grew up... |
| PRUNE (aligned) | Eugen Roth's work office is surrounded by the streets of Paris. He has 3 different routes to get to his office from his home. He takes the first route for 10 minutes, the second route for 15 minutes, and the third route for 20 minutes. If he decides to use each route once every week, how many minutes does he spend traveling in total over the week? To determine the total time Eugen Roth spends traveling to his office over the week... |
| RECT (aligned) | Eugen Roth's work office is surrounded by Paris's most exclusive neighborhoods, which are full of high-end shops and restaurants. If the office is located on the 6th floor of a building, and the building's floors are numbered starting from the ground floor, how many floors are there above Eugen's office? To determine how many floors are above Eugen Roth's office, we need to consider the following information: 1. The office is located on the 6th floor... |

