# OpenReview forum: "The Labyrinth and the Thread: Rethinking Regularizations in Sequential Knowledge Editing for Large Language Models"
_ICML.cc/2026/Conference — ICML 2026 regular_

### Official Review · Reviewer_geXz · 2026-02-22

**Soundness:** 3
**Presentation:** 3
**Significance:** 3
**Originality:** 3
**Overall Recommendation:** 5
**Confidence:** 3

**Summary:**

This paper provides a theoretical unification of sequential knowledge editing and one-time editing in LLMs. It also illustrates that the actual success of AlphaEdit comes from implicitly maintaining OTE-SE equivalence, not by the null-space projection.

**Compliance With Llm Reviewing Policy:**

Affirmed.

**Final Justification:**

The authors have provided extensive expriments that addresses my most concerns.

**Key Questions For Authors:**

1. The empirical proof of the equivalence is only done on one set of hyperparameters, while it could be more convincing if it's validated in different settings.

**Limitations:**

yes

**Strengths And Weaknesses:**

Strengths:
1. The paper presents rigorous theoretical proof, which makes it more convincable.
2. The equivalence of one-time and sequential editing is interesting and could be potentially helpful
3. The paper is well-written with clear narrative

Weakness:
1. Only locate-and-edit (parameter-modifying) methods are analyzed.
2. The insight of equivalence of one-time and sequential editing is still limitd as it does not provide evidence on how existing knowledge could be preserved, since edit all knowledge at once could also greatly harm the performance.

---

> ### Author Rebuttal · Authors · 2026-03-31
>
> We thank the reviewer for the positive assessment. Below we address each concern.
>
> ## W1. Only locate-and-edit (parameter-modifying) methods are analyzed.
>
> We acknowledge this limitation: our theory does not cover parameter-preserving paradigms as they usually introduce non-linear modules that break the OLS framework. Still, OLS-based locate-and-edit is the dominant direct weight-editing paradigm and lacked a unified stability theory until now. The impact also extends further:
>
> 1. **Broad applicability.** OLS-based editing is used beyond knowledge editing—AlphaSteer [1] for safety, AnyEdit [2] for reasoning, and [3] documents potential OLS interventions across multilingualism, persona control, and more. Our OTE–SE insight transfers to *any* sequential OLS setting.
>
> 2. **Forward-looking design recipe:** Remark 3.3/Prop. 3.4 give a constructive procedure to derive stable SE from *any* OTE objective with shifted quadratic representability—any future OLS method immediately obtains a provably stable SE variant.
>
> 3. **Novel algorithms:** We present (a) Algorithm 1 for error correction of post-processing regularization; (b) Memorize-the-Latest (Remark A.4)—a new fully-conflicting SE capability, validated on four LLMs (Tables 7–8).
>
> ### References:
> [1] Sheng et al., "AlphaSteer: Learning Refusal Steering with Principled Null-Space Constraint," ICLR 2026.
>
> [2] Jiang et al., "AnyEdit: Edit Any Knowledge Encoded in Language Models," ICML 2025.
>
> [3] Zhang et al. "Locate, Steer, and Improve: A Practical Survey of Actionable Mechanistic Interpretability in Large Language Models," arXiv preprint arXiv:2601.14004 (2026).
>
>
> ## W2. No evidence that existing knowledge is preserved.
>
> We conducted a **Wikipedia drift analysis** after 2,000 SE edits, measuring PPL and top-1 token agreement:
>
> **LLaMA-3 (PPL 9.27):**
>
> | Method | PPL | Top-1 Agree |
> |---|---|---|
> | RECT (aligned) | 9.52 | 0.926 |
> | MEMIT (aligned) | 9.70 | 0.904 |
> | RECT (naïve) | 303,783 | 0.001 |
> | MEMIT (naïve) | 54,062,861 | 0.000 |
>
> **Qwen2.5:** OTE-aligned PPL 9.28–9.39 (baseline 9.15), agreement >93%; naïve methods degrade to PPL≈111, agreement≈53%.
>
> OTE-aligned SE preserves existing knowledge; naïve SE leads to catastrophic collapse. Together with GLUE evaluations (Figure 2 & 5), this confirms OTE–SE alignment is critical for knowledge preservation. Full results will be added in revision.
>
> ## Q1. OTE-SE Equivalence validated on only one hyperparameter setting.
>
> We provide three lines of evidence:
>
> ### 10,000 edits (5x editing steps).
>
> | Algorithm | Model | Eff. | Gen. | Spe. | Flu. | Consist. |
> |-|-|-|-|-|-|-|
> | MEMIT (aligned) | LLaMA-3 | 98.53 | 91.06 | 61.42 | 604.44 | 32.32 |
> | RECT (aligned) | LLaMA-3 | 98.88 | 91.22 | 64.76 | 620.46 | 32.16 |
> | MEMIT (aligned) | Qwen2.5 | 99.59 | 85.42 | 75.68 | 624.07 | 30.49 |
> | RECT (aligned) | Qwen2.5 | 99.47 | 84.77 | 76.57 | 624.17 | 30.66 |
> | MEMIT (aligned) | GPT-J | 99.35 | 93.73 | 64.44 | 612.36 | 40.24 |
> | RECT (aligned) | GPT-J | 99.33 | 93.91 | 66.93 | 615.34 | 41.02 |
> | MEMIT (aligned) | GPT2-XL | 91.23 | 78.28 | 56.34 | 545.65 | 26.44 |
> | RECT (aligned) | GPT2-XL | 93.71 | 81.16 | 58.11 | 539.44 | 27.25 |
>
> Even at 10K edits, efficacy remains >91% and generalization >78% across all models.
>
> **GLUE:** No catastrophic drop at 10K edits—avg F1 drops modestly from 0.627→0.555 (LLaMA-3) and 0.673→0.647 (Qwen2.5). The model remains fully functional, in stark contrast to naïve methods which collapse (PPL>300K) well before this scale.
>
> **Example Generation:** Danielle Darrieux was born in London, England, to a Danish mother and an American father. Her mother was an actress...
>
> ### Different task formulation.
> OTE–SE equivalence also holds under **Memorize-the-Latest** (Section 3.1, Remark A.4), where each step fully contradicts all previous edits. Stable results across four LLMs (Tables 7–8) confirm the principle generalizes beyond standard editing.
>
> ### Borader than just hyperparameters
>
> We emphasize that the paper's experiments already span a **combinatorial grid** of the key algorithmic choice beyond mere hyperparameters. It includes discussions of $\lambda$ (L2 regularization strength), $P$ (projection matrix), and $\mathcal{R}_p$ (post-processing operator)—through the four editing methods:
>
> | Method | $\lambda$ | $P$ | $\mathcal{R}_p$ |
> |-|-|-|-|
> | MEMIT (aligned) | 0 | $I$ | — |
> | AlphaEdit (aligned) | 10 | Null-space proj. | — |
> | PRUNE (aligned) | — | — | Eigenvalue pruning |
> | RECT (aligned) | — | — | Top-50% truncation |
> | RECT-20% (aligned) | — | — | Top-20% truncation (Table 9) |
>
> OTE–SE equivalence holds across **all** configurations (Tables 4–6, Figures 2–3). This consistency is exactly what Proposition 3.2 guarantees for any $\lambda \ge 0$, any $P$, and any regularization choices.

---

> > ### Author Rebuttal · Reviewer_geXz · 2026-04-02
> >
> > I appreciate the response of the authors, I will raise my score

---

> > > ### Author Response · Authors · 2026-04-03
> > >
> > > Dear Reviewer geXz,
> > >
> > > Thank you very much for your encouraging comments and for raising your score — we truly appreciate it. Your constructive feedback has helped us improve the paper, and we are deeply grateful for your time and expertise.
> > >
> > > We also appreciate your acknowledgment that our response has partially resolved your concerns. In case you have further questions, we would be very happy to clarify any points or provide further revisions based on your suggestions.
> > >
> > > Thank you again for your thoughtful engagement with our work. We look forward to your further guidance.
> > >
> > > Best regards,
> > >
> > > The Authors

---

### Official Review · Reviewer_2vM8 · 2026-03-08

**Soundness:** 3
**Presentation:** 3
**Significance:** 2
**Originality:** 3
**Overall Recommendation:** 4
**Confidence:** 3

**Summary:**

The paper investigates the mechanisms of stable sequential knowledge editing in Large Language Models, identifying a formal mathematical equivalence between one-time and sequential editing as the core driver of stability. By demonstrating that many complex regularization strategies are redundant if past editing constraints are properly incorporated, the authors provide a unified theoretical framework and error-correction algorithms that ensure consistent model performance across thousands of updates without sacrificing general linguistic capabilities.

**Compliance With Llm Reviewing Policy:**

Affirmed.

**Final Justification:**

I appreciate the authors’ response and the clarifications provided. I have decided to raise my Overall Recommendation from 3 to 4.

**Key Questions For Authors:**

See weaknesses.

**Limitations:**

The experimental evaluation currently considers up to 2,000 edits. Additional experiments under more extreme lifelong editing settings (e.g., 10,000 edits) would further strengthen the evidence that the proposed method maintains stability without model collapse.

**Strengths And Weaknesses:**

Strengthes:

1. A key strength of this work is its critical re-evaluation of the null-space projection in AlphaEdit. By introducing the "Memorize-the-Latest" task, the authors demonstrate that null-space constraints are not the fundamental factor for stability and actually introduce accumulation errors. This discovery shifts the focus toward OTE-SE equivalence as the true mechanism governing successful sequential updates.

2.The paper provides comprehensive and detailed experimental validation across four large language models: GPT-2 XL, GPT-J, LLaMA-3, and Qwen-2.5. The evaluation utilizing multiple benchmarks and GLUE tasks to validate that general linguistic capabilities are preserved after thousands of sequential edits.


Weaknesses:

1. Insufficient evaluation in extreme scenarios: The experimental setup is not rigorous enough compared to prior work, as the current evaluation only tests up to 2,000 total edit samples. In contrast, the original MEMIT paper evaluated batch sizes of 10,000, and recent studies [1,2] demonstrate performance under significantly more extreme lifelong editing conditions. To fully validate the effectiveness of the proposed OTE-SE equivalence, the authors should provide comparative results for sequential editing with a total volume of at least 10,000 edits.

2. Limited Novelty in Core Motivation: The central motivation of the paper, namely that sequential editing should remain consistent with a global one-time editing objective, appears conceptually related to the formulation in SimIE [2]. Both works are based on the idea that instability in lifelong editing arises because sequential updates drift away from an idealized global solution. While this paper offers a more formal derivation of the OTE-SE equivalence and develops the argument in a mathematically cleaner way, the underlying intuition is not entirely new. The manuscript would be strengthened by a clearer discussion of how its theoretical contribution differs from, and goes beyond, the “Ideal Editor” framework introduced in SimIE.

[1]LyapLock: Bounded Knowledge Preservation in Sequential Large Language Model Editing

[2]Towards Lifelong Model Editing via Simulating Ideal Editor

---

> ### Author Rebuttal · Authors · 2026-03-31
>
> ## Q1: Insufficient evaluation in extreme scenarios
>
> We appreciate this concern. We provide additional experiments at **10,000 edits** (5× the original scale) across all four models.
>
> **Editing performance at 10K edits.**
>
> | Algorithm | Model | Eff. | Gen. | Spe. | Flu. | Consist. |
> |-|-|-|-|-|-|-|
> | MEMIT (aligned) | LLaMA-3 | 98.53 | 91.06 | 61.42 | 604.44 | 32.32 |
> | RECT (aligned) | LLaMA-3 | 98.88 | 91.22 | 64.76 | 620.46 | 32.16 |
> | MEMIT (aligned) | Qwen2.5 | 99.59 | 85.42 | 75.68 | 624.07 | 30.49 |
> | RECT (aligned) | Qwen2.5 | 99.47 | 84.77 | 76.57 | 624.17 | 30.66 |
> | MEMIT (aligned) | GPT-J | 99.35 | 93.73 | 64.44 | 612.36 | 40.24 |
> | RECT (aligned) | GPT-J | 99.33 | 93.91 | 66.93 | 615.34 | 41.02 |
> | MEMIT (aligned) | GPT2-XL | 91.23 | 78.28 | 56.34 | 545.65 | 26.44 |
> | RECT (aligned) | GPT2-XL | 93.71 | 81.16 | 58.11 | 539.44 | 27.25 |
>
> Efficacy remains >91% and generalization >78% across all models at 10K edits.
>
> **General capability preservation (GLUE) at 10K edits.**
>
> | | SST | MMLU | MRPC | CoLA | RTE | NLI | AVG |
> |-|-|-|-|-|-|-|-|
> | Qwen2.5 Base | 0.825 | 0.637 | 0.729 | 0.810 | 0.226 | 0.809 | 0.673 |
> | Qwen2.5 10K | 0.721 | 0.644 | 0.724 | 0.785 | 0.219 | 0.790 | 0.647 |
> | LLaMA-3 Base | 0.831 | 0.562 | 0.658 | 0.761 | 0.284 | 0.666 | 0.627 |
> | LLaMA-3 10K | 0.782 | 0.360 | 0.690 | 0.647 | 0.319 | 0.531 | 0.555 |
>
>
> Qwen2.5 shows only marginal degradation (avg 0.673→0.647). LLaMA-3 exhibits a larger drop (0.627→0.555), primarily driven by MMLU, which broadly probes world knowledge and is expectedly more sensitive to large-scale factual rewrites. This moderate, model-dependent degradation is consistent with prior work (e.g., LyapLock [1] shows up to 20% degradation on MRPC and ~10% on MMLU at comparable/smaller scales in their Figure 3). Importantly, as shown in Figure 5 of our paper, naive (non-aligned) SE methods suffer complete GLUE collapse to near zero *within 2,000 edits*, underscoring that OTE-SE alignment is the essential mechanism preventing catastrophic degradation. The model also remains fully functional in generation after 10K edits (e.g., *"Danielle Darrieux was born in London, England, to a Danish mother and an American father. Her mother was an actress..."*).
>
> ## Q2: Novelty over SimIE & Additional Related Work
>
> ### Related Work:
> We thank the reviewer for raising LyapLock [1] and SimIE [2]; we will add a dedicated discussion in Section 5 (Related Works).
>
> ### SimIE and Our Work:
> Although both works share the intuition that sequential editing should stay anchored to a global objective, our contributions are fundamentally different. SimIE proposes *a heuristic to approximate an ideal editor*; in contrast, our work (1) *proves* that exactly following the ideal editor is achievable without approximation, (2) *diagnoses why* certain methods succeed or fail — insights entirely absent from SimIE, and (3) *generalizes* the framework to broader SE objectives beyond reproducing one-time edits (e.g., memorize-the-latest).
>
> **1. Exact equivalence vs. heuristic approximation.**
> SimIE's recursive correction relies on over-parameterization and key-value invariance assumptions. Our Prop. 3.4 establishes *exact* OTE-SE equivalence for any shifted-quadratic objective. SimIE's objective $\||SK-V\||^2+\lambda\||S\||^2$ is a special case; Remark 3.3 directly yields the exact update $\Delta^*_t = r_t k_t^\top P_t^{-1}$, eliminating SimIE's approximation error. __Empirically__, OTE-SE alignment maintains >98% efficacy at large scale, vs. <80% for MEMIT+SimIE (SimIE Table 1).
>
> **2. Diagnosing why methods succeed or fail — beyond SimIE's scope.**
> Our work provides two diagnostic insights absent from SimIE: (i) AlphaEdit's stability stems not from null-space projection but from inadvertently preserving OTE-SE equivalence (Section 3.1, Table 1), revising the prevailing understanding; (ii) post-processing regularizers (PRUNE, RECT) break equivalence via accumulating per-step deviations — a problem SimIE never dicussed. Algorithm 1 is the first error-correction solution for this class, and we further show many regularization strategies become nonessential once OTE-SE alignment holds. These insights help predicting failure of editing methods without expensive experiments.
>
> **3. Editor-agnostic generality across objectives.**
> SimIE uses a sequential algorithm to reproduce step-wise one-time edit results. However, sequential editing is inherently richer than merely applying one-time edits one after another. Our framework goes beyond this paradigm: Prop. 3.4 applies to *any* OTE objective with shifted quadratic representability, not only covering standard one-time editors (MEMIT, AlphaEdit, PRUNE, RECT, and SimIE itself) but also enabling SE algorithms for fundamentally different editing goals (e.g., memorize-the-latest retains only the current fact — a task SimIE's formulation cannot express).
>
> We will add a discussion in Section 5 and a remark in Section 3.2 to show these distinctions.

---

> > ### Author Rebuttal · Reviewer_2vM8 · 2026-04-02
> >
> > Thank you to the authors for their response and for addressing my concerns. I decide to raise my Overall Recommendation from 3 to 4.

---

> > > ### Author Response · Authors · 2026-04-03
> > >
> > > Dear Reviewer 2vM8,
> > >
> > > Thank you very much for your encouraging feedback and for raising your Overall Recommendation from 3 to 4. We truly appreciate the time and care you’ve taken in reviewing our work.
> > >
> > > We are glad to hear that your concerns have been fully resolved and that our revisions adequately addressed your comments. Your insights were invaluable in helping us improve the paper.
> > >
> > > Thank you again for your support and constructive guidance.
> > >
> > >
> > > Best regards,
> > >
> > > The Authors

---

### Official Review · Reviewer_1yig · 2026-03-11

**Soundness:** 3
**Presentation:** 3
**Significance:** 3
**Originality:** 2
**Overall Recommendation:** 4
**Confidence:** 3

**Summary:**

This paper studies a fundamental question in sequential knowledge editing: what truly makes sequential editing stable over many updates. Rather than treating regularization or null-space projection as the main reason behind the success of prior methods such as AlphaEdit, the paper argues that the key factor is the equivalence between sequential editing and its corresponding one-time editing objective. Building on this view, the authors provide a theoretical analysis showing that the stability of sequential editing comes from preserving this OTE-SE equivalence, reinterpret AlphaEdit from this perspective, and further extend the framework to regularized methods such as PRUNE and RECT through error correction.

**Compliance With Llm Reviewing Policy:**

Affirmed.

**Final Justification:**

The rebuttal has adequately addressed my concerns, and I am happy to maintain my positive score.

**Key Questions For Authors:**

Same as the weaknesses above.

**Limitations:**

Yes.

**Strengths And Weaknesses:**

## Strengths
- I think the main conceptual contribution is strong. The paper does not simply introduce another editing variant, but instead provides a deeper explanation of why some sequential editing methods work well. The OTE-SE equivalence perspective is clean, intuitive in hindsight, and potentially useful as a design principle for future methods.

- I found the reinterpretation of AlphaEdit particularly valuable. The paper makes a convincing case that the benefit does not fundamentally come from null-space projection itself, but from preserving equivalence to the corresponding one-time editing solution. This helps clarify a point that could otherwise be misunderstood in the literature.

- I appreciate that the paper goes beyond one specific method and gives a more unified framework covering MEMIT, AlphaEdit, PRUNE, and RECT. This makes the work more broadly relevant than a method-specific analysis.

## Weaknesses
- One question I had concerns the analysis around Eq. (4) and Table 1. The paper argues that AlphaEdit’s null-space simplification becomes problematic because an error term is ignored in the derivation, which eventually harms stability and general capability. This explanation is interesting and convincing at a conceptual level. At the same time, I was curious whether the authors have directly verified the complementary side of this claim: if that omitted error term is explicitly retained or corrected for in AlphaEdit, would the sequential editing performance improve accordingly? Even a small additional experiment or discussion here could make the mechanism interpretation even more complete.

- Relatedly, since the main message of the paper is that OTE-SE equivalence is more fundamental than the specific regularization design, I think it would be very useful to isolate this factor more explicitly. For example, have the authors tested whether MEMIT or AlphaEdit with only OTE-SE alignment, but without any additional regularization, can still maintain strong performance under long sequential editing? Such an ablation would further strengthen the claim that the alignment principle itself is the key ingredient.

---

> ### Author Rebuttal · Authors · 2026-03-31
>
> We thank the reviewer for the constructive feedback and for recognizing the OTE–SE equivalence perspective and the unified framework. We address each question below.
>
> ## W1: Direct verification that retaining the omitted error term improves performance
>
> Yes—we provide evidence from three angles:
>
> **1. Table 1 (Section 3.1)** directly compares the "Full Version" (retaining the error term) vs. the "Null Space" variant (omitting it) on GLUE after sequential editing of LLaMA-3. The Null-Space variant collapses to near-zero on all six tasks, while the Full Version preserves pre-edit performance. This confirms that retaining the omitted term restores stability.
>
> **2. New result: Memorize-the-Latest with AlphaEdit's full formulation.** We further evaluate AlphaEdit's full objective on the Memorize-the-Latest task (with $P$ but without discarding the error term):
>
>
> Forgotten knowledge: **lower Eff., Gen.** values are better (should return to pre-edit state)
> | **LLaMA-3** | Eff. | Gen. | Spe. | Flu. | Consis. |
> |---|---|---|---|---|---|
> | Pre-edited | 7.85 | 10.58 | 89.48 | 635.23 | 24.14 |
> | Forgotten Knowledge (Null Space) | 54.16 | 52.92 | 50.09 | 551.13 | 1.74 |
> | Forgotten Knowledge (Full Ver.) | 15.42 | 15.24 | 88.51 | 633.71 | 24.09 |
>
> Latest Knowledge: **higher Eff., Gen.** values are better (should memorize these knowledge)
> | **LLaMA-3** | Eff. | Gen. | Spe. | Flu. | Consis. |
> |---|---|---|---|---|---|
> | Pre-edited | 7.85 | 10.58 | 89.48 | 635.23 | 24.14 |
> | Latest Knowledge (Null Space) | 92.0 | 79.0 | 45.8 | 530.72 | 7.43 |
> | Latest Knowledge (Full Ver.) | 100.0 | 98.0 | 88.2 | 631.1 | 32.55 |
>
> **The full version:** the latest knowledge achieves 100% efficacy and 98% generalization; forgotten knowledge returns to pre-edit baselines; fluency/consistency are preserved.
>
> **Null Space Version:** dropping null-space terms leads to much worse editing performance -- less efficacy on the latest knowledge and worse forgetting performance (model remains far from pre-edit states). We also observe much worse fluency/consistency, aligned with GLUE results in Table 1 and generation example (`Karl Lachmann was born in Berlin ( Canada ( ( ( ( Toronto...`).
>
> **3. Tables 7–8 (Appendix C.5)** validate this across all four LLMs under our conflict-resolution framework. The extreme case of a full conflicting sequence is the memorize-the-latest task you mentioned here (details in Remark A.4). We empirically get ≥99.67% efficacy and ≥93% generalization on the latest knowledge throughout. We will add cross-references in Section 3.4 for better clarity.
>
>
> ## W2: Ablation isolating OTE–SE alignment without any regularization
>
> The paper already contains precisely this important case. Setting $P = I \text{ and } \lambda = 0$ in Proposition 2 recovers exactly the **MEMIT (aligned)** algorithm, with update rule in Remark A.1. Thus, **MEMIT (aligned) is the OTE–SE aligned method with no regularization whatsoever**—no null-space projection, no post-processing, no $L^2$ penalty.
>
> As shown in our results:
> - **MEMIT (aligned) matches or exceeds regularized methods** (AlphaEdit, PRUNE, RECT) across all four LLMs under 2,000 sequential edits (Table 2 / Figure 2, Table 3).
> - **General capabilities are fully preserved** (Figure 3, GLUE benchmark).
> - **The ablation (Tables 4–5)** shows that "Not OTE Aligned" (Naive) variants of *all* methods—including regularized ones—collapse (efficacy ~99% → ~56%), while OTE-aligned variants remain stable regardless of regularization.
> - **Additional results with 10K edits.** We also show that even with 10K edit, the MEMIT (aligned) method without regularizations works stably and achieve strong editing performance, as shown in the table below:
>
> | Algorithm | Model | Eff. | Gen. | Spe. | Flu. | Consist. |
> |-|-|-|-|-|-|-|
> | MEMIT (aligned) | LLaMA-3 | 98.53 | 91.06 | 61.42 | 604.44 | 32.32 |
> | MEMIT (aligned) | Qwen2.5 | 99.59 | 85.42 | 75.68 | 624.07 | 30.49 |
> | MEMIT (aligned) | GPT-J | 99.35 | 93.73 | 64.44 | 612.36 | 40.24 |
> | MEMIT (aligned) | GPT2-XL | 91.23 | 78.28 | 56.34 | 545.65 | 26.44 |
>
> This confirms that **OTE–SE alignment alone is sufficient**; regularization is neither necessary nor sufficient for stability. We will highlight MEMIT (aligned) as the "no-regularization" baseline explicitly in Section 4.1.

---

> > ### Author Rebuttal · Reviewer_1yig · 2026-04-02
> >
> > Thank you to the authors for the rebuttal. They have addressed most of my concerns well, and I am willing to maintain my positive score.

---

> > > ### Author Response · Authors · 2026-04-03
> > >
> > > Dear Reviewer 1yig,
> > >
> > > Thank you very much for your thoughtful feedback and for acknowledging the revisions in our rebuttal.
> > >
> > > We are truly glad to hear that your concerns have been fully resolved and that you are willing to maintain your positive score. Your constructive comments have helped us improve the paper, and we greatly appreciate your time and expertise.
> > >
> > > Best regards,
> > >
> > > The Authors

---

### Official Review · Reviewer_QdGG · 2026-03-12

**Soundness:** 3
**Presentation:** 3
**Significance:** 3
**Originality:** 3
**Overall Recommendation:** 4
**Confidence:** 4

**Summary:**

This paper studies the foundations of stable sequential knowledge editing for large language models. The main claim is that the stability of sequential editing does not fundamentally come from specialized regularization or null-space projection, but from preserving equivalence between one-time editing (OTE) and sequential editing (SE). The paper unifies a class of locate-and-edit methods under an OLS-based perspective, proves equivalence results for AlphaEdit-style formulations, proposes a general principle for constructing stable sequential updates, and introduces an error-correction view for post-processing regularization methods. The paper also discusses a formulation for handling conflicting edits. Experiments on multiple LLMs and standard knowledge-editing benchmarks compare aligned and naive variants of MEMIT, AlphaEdit, PRUNE, and RECT, showing that OTE-aligned sequential updates are substantially more stable than non-aligned alternatives.

**Compliance With Llm Reviewing Policy:**

Affirmed.

**Key Questions For Authors:**

1. Scope of the theoretical framework:
The central theoretical claims are derived under an OLS-style formulation and related quadratic assumptions. Could the authors clarify more explicitly which classes of model-editing methods are covered by the theory, and which important editing methods fall outside this framework? A clearer boundary of applicability would strengthen my confidence in the generality of the paper’s conclusions.


2. Empirical validation of conflict resolution:
The paper presents a general sequential editing update with conflict resolution, which seems potentially important in practice. However, this contribution is not strongly validated in the main experiments. Could the authors provide empirical results on genuinely conflicting or contradictory edit streams? Such results would help assess whether this extension is practically useful rather than mainly theoretical.

3. Broader side-effect evaluation:
The experiments focus primarily on standard editing metrics such as efficacy, generalization, and specificity. Did the authors evaluate broader downstream side effects or behavior shifts beyond these metrics, especially after long editing sequences?


4. Role of regularization beyond the analyzed family:
The paper argues that stability is driven primarily by OTE-SE equivalence rather than specialized regularization. Could the authors clarify whether they believe this conclusion holds only within the locate-and-edit / OLS family, or more broadly across other model-editing paradigms as well?

**Limitations:**

The paper includes an Impact Statement, but the discussion of limitations and potential risks could be improved. In particular, the manuscript would benefit from a more explicit discussion of the scope of the theoretical assumptions, the practical limitations of the conflict-resolution formulation, and the risks of deploying imperfect sequential editing methods in safety-critical or misinformation-sensitive settings.

**Strengths And Weaknesses:**

-) Strengths:

1. Strong conceptual contribution:

The paper provides a clear and potentially important conceptual simplification of sequential knowledge editing by arguing that OTE-SE equivalence is the key principle underlying stable updates, rather than various specialized regularization mechanisms.

2. Non-trivial theoretical analysis:

The paper contains meaningful theoretical development, including the equivalence result for AlphaEdit-style updates, a generalized formulation for sequential construction from OTE objectives, and a principled view of error correction for post-processing regularization. This is stronger than many purely empirical model-editing papers.


3. Useful unification of prior methods:

A notable strength is that the paper does not merely propose another editing variant, but instead offers a unifying perspective that helps explain why AlphaEdit, MEMIT, PRUNE, and RECT behave differently under repeated edits. This kind of synthesis is valuable for the model editing community.

4. Empirical validation across multiple models and datasets:

The experiments cover four base models and two standard editing benchmarks, and include aligned vs. naive ablations. This gives reasonably broad empirical support for the main claims.


5. Practical relevance:

If correct, the proposed perspective could simplify the design of future lifelong editing algorithms and reduce reliance on complicated regularization heuristics. That makes the work both theoretically and practically relevant.

...................................................................................

-) Weaknesses:

1. Scope of the theoretical claims may be narrower than the paper sometimes suggests:

The core theory seems to rely on a particular OLS-style formulation and shifted quadratic assumptions. It is not fully clear how far the conclusions extend beyond the class of locate-and-edit methods that admit this structure. As written, some claims about the unimportance of regularization may sound broader than what is actually proven.


2. Limited empirical evaluation of the conflict-resolution extension:

The paper presents a general formulation for conflicting edits, which is potentially interesting, but this part appears largely theoretical and deferred to the appendix, without strong empirical validation in the main experiments. As a result, one of the advertised contributions is less convincing than the rest.


3. Evaluation focuses mostly on standard editing metrics, with limited broader downstream analysis:

The paper evaluates efficacy, generalization, and specificity on CounterFact/ZsRE, and includes some language capability analysis, but the broader impact on general behavior is still somewhat limited in scope. More extensive evaluation of unintended side effects would strengthen the practical claims.

4. Some of the empirical gains come from theory-aligned reformulations of prior methods rather than a fully distinct new editing algorithm:

This is not a fatal issue, but it means the contribution is more of a theoretical reinterpretation and framework for reconstruction than a clearly separate end-to-end editing method. Some readers may view this as reducing methodological novelty.

---

> ### Author Rebuttal · Authors · 2026-03-31
>
> We thank the reviewer for the positive assessment. Below we address each concern:
>
> ## Q1 & Q4. Scope / Role of Regularization
>
> We clarify the scope below. **In scope:** Locate-and-edit methods with OLS-based updates (MEMIT, AlphaEdit, PRUNE, RECT), and any method satisfying *shifted quadratic representability* (Prop. 3.4). OTE–SE equivalence holds for any projection $P$ and $\lambda \ge0$. **Out of scope:** Methods with non-linear updates — such as parameter-preserving editors (GRACE), meta-learning editors (MEND), and fine-tuning approaches.
>
> **On regularization (Q4):** Within the OLS family, stability is a *structural* property of properly OTE-SE alignment, not of any regularizer. The intuition may help other application domains using OLS to obtian model modifications *but we do not claim formal results beyond the OLS family*.
>
> **Possible Broader impact:** OLS locate-and-edit is increasingly used beyond knowledge editing: AlphaSteer [1] uses OLS for safety steering vectors; AnyEdit [2] extends it to reasoning tasks. [3] also identifies potential OLS interventions across multilingualism, safety, persona control, and reasoning. Our OTE–SE insight transfers to any sequential OLS setting.
>
>
> ### Reference
> [1] Sheng et al., "AlphaSteer: Learning Refusal Steering with Principled Null-Space Constraint," ICLR 2026.
>
> [2] Jiang et al., "AnyEdit: Edit Any Knowledge Encoded in Language Models," ICML 2025.
>
> [3] Zhang et al. "Locate, Steer, and Improve: A Practical Survey of Actionable Mechanistic Interpretability in Large Language Models," arXiv preprint arXiv:2601.14004 (2026).
>
> ## Q2. Conflict Resolution Validation
>
> We have in fact already validated conflict resolution empirically. Our **Memorize-the-Latest** task is an extreme conflict resolving case:  every step fully contradicts the previous one, and at each step, the model retains only the latest edit while discarding all prior edits. Formally (Remark A.4), this is a special case of our conflict-resolution formula (Prop. 3.5).
>
> **Results (Tables 7–8, Appendix C.5):** Across four LLMs, latest knowledge achieves ≥99.67% efficacy and ≥93% generalization; forgotten knowledge returns to pre-edit baselines; fluency/consistency are preserved. This validates conflict resolution under extreme conditions. We will add a cross-reference in Section 3.4 for better clarity.
>
> ## Q3. Broader Side-Effect Evaluation
>
> We note that the latent visualizations (Figure 4) can help show general distribution shift in the latent space is small. We further conducted drift analysis on **Wikipedia**, measuring PPL and top-1 token agreement after 2000 edits:
>
> **LLaMA-3 (PPL 9.27):**
>
> | Method | PPL_edit | Top-1 Agree |
> |---|---|---|
> | RECT (aligned) | 9.52 | 0.926 |
> | MEMIT (aligned) | 9.70 | 0.904 |
> | RECT (naïve) | 303,783 | 0.001 |
> | MEMIT (naïve) | 54,062,861 | 0.000 |
>
> Similar trends hold on **Qwen2.5**: OTE-aligned PPL 9.15→9.28–9.39, agree >93%; naïve methods degrade to PPL=111, agree=53%. OTE–SE alignment is essential for preserving broad model behavior. We will include full results in the revision.
>
> ## Q5. On Methodological Novelty: Theory-Driven Framework as a Contribution
>
> We agree the primary contribution is a theoretical framework rather than a new end-to-end editing method. We kindly argue this is the right contribution to the growing heuristics lacks principled understanding. Concretely, our framework delivers:
>
> 1. **Explanatory power:** The first rigorous explanation for why AlphaEdit succeeds and naïve SE fails.
>
> 2. **Forward-looking design recipe:** Remark 3.3/Prop. 3.4 give a constructive procedure to derive stable SE from *any* OTE objective with shifted quadratic representability—any future OLS method immediately obtains a provably stable SE variant.
>
> 3. **Concrete algorithms:** Algorithm 1 for error correction of post-processing regularization, and Memorize-the-Latest (Remark A.4) — a new capability for fully-conflicting SE, validated across four LLMs (Tables 7–8).
>
> ## Q6. Clearer Limitations
>
> We thank the reviewer and will expand the limitations discussion to cover three aspects:
>
> 1. **Theoretical scope:** As clarified in Q1, our guarantees apply to the OLS-based family. Methods with non-linear update mechanisms (e.g., GRACE, MEND, fine-tuning) fall outside our formal analysis, though the conceptual principle of OTE–SE alignment may still serve as a useful design heuristic.
>
> 2. **Safety and deployment risks:** Model editing methods could be misused to inject misinformation or subtly alter model behavior. Like all editing methods, even with OTE–SE alignment, edits can still slightly impact LLM general behavior over very long editing horizons. We recommend that safety-critical deployments pair sequential editing with independent verification of model outputs.

---

### Decision · Program_Chairs · 2026-04-30

**Decision:**

Accept (regular)

**Comment:**

- This paper offers a unified perspective that explains the behavior of existing editing methods in sequential editing setting, especially rethinking the null space projection.

- A novel finding is that editing stability emerges naturally from properly accounting for accumulated editing constraints, rather than from specialized regularization or null-space operations, which is quite different from the SOTA methods.

- The proposed solution may have practical impact and could simplify the design of future lifelong editing algorithms.

- Reviewers also have minor evaluation concerns, which have been partially addressed during rebuttal and should be included in the revision.